# OmniSIFT: Modality-Asymmetric Token Compression for Efficient Omni-modal Large Language Models

Yue Ding [1 2 *]  Yiyan Ji [3 *]  Jungang Li [4]  Xuyang Liu [5]  Xinlong Chen [1]  Junfei Wu [1]  Bozhou Li [6]
Bohan Zeng [6]  Yang Shi [6]  Yushuo Guan [2]  Yuanxing Zhang [2]  Jiaheng Liu [3]  Qiang Liu [1]  Pengfei Wan [2]
Liang Wang [1]

## Abstract

Omni-modal Large Language Models (Omni-LLMs) have demonstrated strong capabilities in audio-video understanding tasks. However, their reliance on long multimodal token sequences leads to substantial computational overhead. Despite this challenge, token compression methods designed for Omni-LLMs remain limited. To bridge this gap, we propose **OmniSIFT** (**Omni**-modal **S**patio-temporal **I**nformed **F**ine-grained **T**oken compression), a modality-asymmetric token compression framework tailored for Omni-LLMs. Specifically, OmniSIFT adopts a two-stage compression strategy: (i) a spatio-temporal video pruning module that removes video redundancy arising from both intra-frame structure and inter-frame overlap, and (ii) a vision-guided audio selection module that filters audio tokens. The entire framework is optimized end-to-end via a differentiable straight-through estimator. Extensive experiments on five representative benchmarks demonstrate the efficacy and robustness of OmniSIFT. Notably, for Qwen2.5-Omni-7B, OmniSIFT introduces only 4.85M parameters while maintaining lower latency than training-free baselines such as OmniZip. With merely 25% of the original token context, OmniSIFT consistently outperforms all compression baselines and even surpasses the performance of the full-token model on several tasks. Code is available at https://github.com/dingyue772/OmniSIFT.

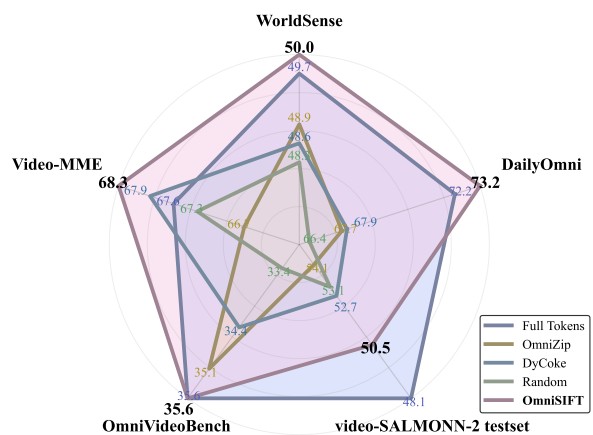

*Figure 1.* **Performance comparison across five audio–video benchmarks**. Results are obtained using Qwen2.5-Omni-7B with a 35% token retained ratio, comparing OmniSIFT against three baseline token compression methods and the full-token baseline.

## 1. Introduction

The rapid evolution of Omni-LLMs (Cheng et al., 2024; Xu et al., 2025b; Liu et al., 2025a) has significantly advanced holistic audio-video-language understanding (Hong et al., 2025; Zhou et al., 2025; Li et al., 2025). However, video signals are composed of densely sampled consecutive frames (Chen et al., 2024b; Jiang et al., 2025a), and audio streams must be encoded at high temporal resolution to capture acoustic dynamics (Ji et al., 2024). When these high-resolution streams are tokenized and interleaved for joint reasoning, the resulting sequence length grows rapidly. For example, a typical 20-second multimodal clip can yield more than 20K tokens (Xu et al., 2025a). Such long token sequences significantly increase computational cost, particularly for long video understanding (Fu et al., 2025).

Token compression (Chen et al., 2024a; Liu et al., 2025d;b; Ye et al., 2025) has emerged as a practical solution to mitigate the prohibitive computational cost caused by excessive token sequences. In the context of vision-centric MLLMs, a

*Equal contribution [1]New Laboratory of Pattern Recognition (NLPR), Institute of Automation, Chinese Academy of Sciences (CASIA) [2]Kling Team, Kuaishou Technology [3]Nanjing University [4]The Hong Kong University of Science and Technology (Guangzhou) [5]Sichuan University [6]Peking University. Correspondence to: Qiang Liu <qiang.liu@nlpr.ia.ac.cn>.

*Proceedings of the 43rd International Conference on Machine Learning*, Seoul, South Korea. PMLR 306, 2026. Copyright 2026 by the author(s).

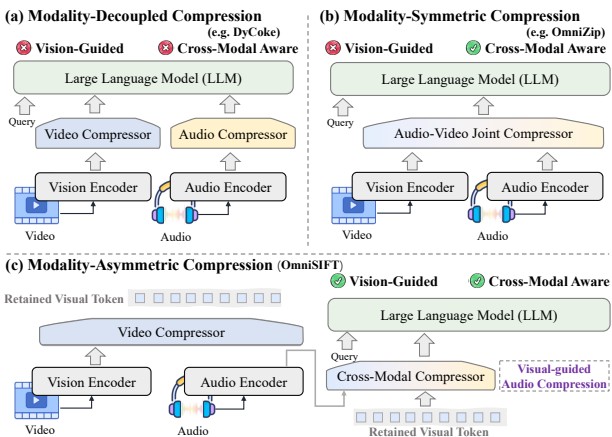

*Figure 2.* **Compression paradigm comparison for Omni-LLMs.** Token compression for Omni-LLMs can be categorized into three paradigms: (a) modality-decoupled compression (left top), which applies audio and video compression independently; (b) modality-symmetric compression (right top), which treats the two modalities equally informative; and (c) modality-asymmetric compression (bottom, ours), which first prunes visual redundancy and then performs visually guided audio compression.

substantial body of work has explored effective strategies for pruning redundant visual tokens (Chen et al., 2024a; Tao et al., 2025b; Yao et al., 2025), demonstrating that significant efficiency gains can be achieved with minimal performance degradation. However, directly extending these approaches to audio–video understanding in Omni-LLMs is far from straightforward. As illustrated in Fig. 2, the modality-decoupled compression method directly transfers vision-only techniques to both video and audio streams. While simple, this strategy completely ignores cross-modal semantic dependencies (Seo et al., 2023) and may discard tokens that are jointly informative.

A recent line of work adopts a modality-symmetric token compression paradigm. OmniZip (Tao et al., 2025c) follows this paradigm by first compressing audio tokens using attention scores from the audio encoder, and then guiding video token pruning with audio-derived saliency. Its reliance on attention-based saliency limits compatibility with efficient operators such as FlashAttention (Shah et al., 2024). In addition, treating the two modalities as equally informative collapses the compression process into selecting salient temporal positions, rather than capturing modality-specific semantic cues. EchoingPixels (Gong et al., 2025) also adopts a modality-symmetric design, performing global cross-modal contextualization over all audio and video tokens via four additional LLM decoder layers before compression. This compression method delays compression to a late stage and introduces substantial computational overhead.

In practice, humans process audio-video content asymmetrically (Koppen et al., 2008). In joint audio-video inputs, redundancy on the video side can often be estimated from the internal structure of the visual stream, whereas the saliency of audio signals is more context dependent and often relies on whether the visual scene provides a semantic anchor (Zhao et al., 2018; Arandjelovic & Zisserman, 2017), such as a visible speaker or a visually grounded event (Chowdhury et al., 2025). This perceptual asymmetry suggests that effective omni-modal token compression should be guided by visual semantics rather than treated symmetrically across modalities.

Taken together, these observations suggest ***three design principles*** for Omni-LLM token compression: (1) Modality-asymmetric, vision-guided compression; (2) Lightweight compression; (3) Compatibility with efficient operators.

Based on the above analysis, we present **OmniSIFT** (**Omni**-modal **S**patio-temporal **I**nformed **F**ine-grained **T**oken compression), a modality-asymmetric framework for visually guided token compression. As illustrated in Figure 2, OmniSIFT first prunes spatial and temporal redundancy in video to produce a compact set of visual anchors, and then uses these anchors to select the audio tokens that are most informative for the scene. This two-stage pipeline removes uninformative signals while preserving the key multimodal cues required for reasoning.

With only 4.85M additional parameters, OmniSIFT achieves lower latency than training-free baselines such as OmniZip on Qwen2.5-Omni-7B. Moreover, with only 25% of the original tokens retained, it consistently outperforms all compression baselines and even surpasses the full-token model on several settings, as illustrated in Figure 1.

Our main contributions are summarized as follows:

- Based on the asymmetric dependency between audio and video, we derive practical design principles for omni-modal token compression.

- We present OmniSIFT, a modality-asymmetric framework that first removes spatial and temporal redundancy in video tokens and then uses the resulting visual anchors to select informative audio tokens.

- Extensive experiments across five benchmarks show that OmniSIFT delivers strong performance–efficiency gains, achieving higher accuracy even with only 25% of the original tokens.

## 2. Related Works

### 2.1. Omni-modal Large Language Models

Omni-LLMs (Jiang et al., 2025b) extend large language models to process heterogeneous modalities within a unified autoregressive framework. Unlike conventional Video-

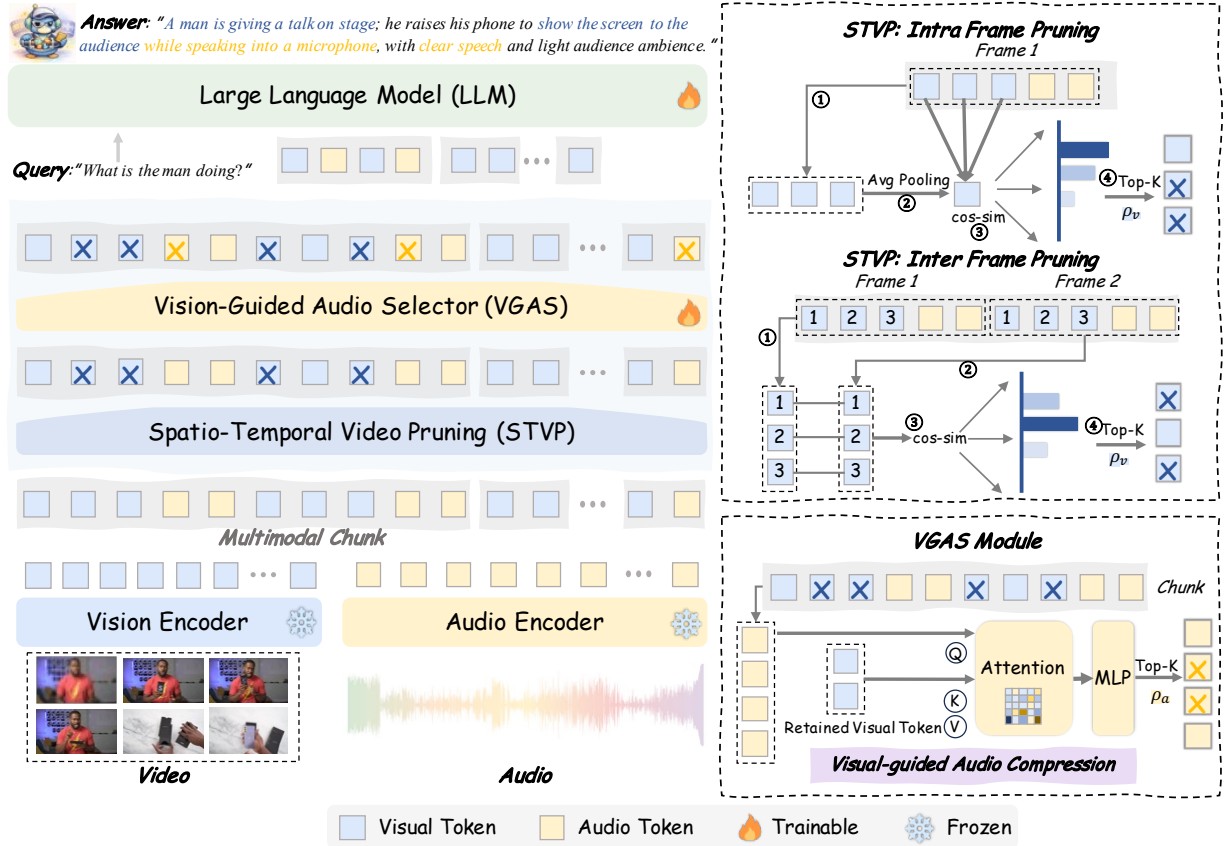

*Figure 3.* **Architecture of OmniSIFT, a modality-asymmetric compression framework**. The framework operates in two stages. In the first stage, STVP removes spatial and temporal redundancy in video tokens to obtain a compact set of visual anchors. In the second stage, VGAS selects audio tokens conditioned on these visual anchors. The resulting compressed multimodal sequence is then fed into the LLM backbone for downstream reasoning.

LLMs (An et al., 2025; Bai et al., 2025; Wu et al., 2025a; Chen et al., 2025c; Wu et al., 2025b), which primarily focus on the interaction between visual sequences and textual instructions (Hu et al., 2023; Li et al., 2026; Xu et al., 2026a;b; Peng et al., 2025), Omni-LLMs additionally incorporate audio signals (Cheng et al., 2024; Tang et al., 2025; Liu et al., 2025a; Chen et al., 2026; Tao et al., 2025a). Proprietary systems such as GPT-4o (Hurst et al., 2024) and Gemini (Comanici et al., 2025) further demonstrate strong performance on audio–visual understanding tasks (Li et al., 2025; Hong et al., 2025; Tao et al., 2026). In the open-source community, models like Qwen2.5-Omni (Xu et al., 2025a) adopt a typical architecture that aligns modality-specific encoders with an LLM through learned projection layers.

## 2.2. Token Compression in Multimodal Models

In the video domain, token compression methods have developed a broad set of mechanisms for reducing visual redundancy. Representative methods estimate token importance through saliency or similarity metrics (Yang et al., 2025; Liu et al., 2025c; Yao et al., 2025; Tao et al., 2025b),

while recent variants explore training-free token filtering and correlation (Han et al., 2026), variation-aware token dropping (Chen et al., 2025a), hierarchical compression for streaming videos (Wang et al., 2025), and curvature-aware spatio-temporal pruning (Lin et al., 2026). These studies also reveal two important considerations for video token reduction: aggressive pruning should preserve holistic context rather than only highlighted tokens (Zou et al., 2026), and video modeling requires balancing temporal gains with spatial preservation (Zhang et al., 2026). However, these methods mainly operate on the visual stream, whereas omni-modal inputs require compression strategies that also preserve cross-modal semantics between video and audio. Along this direction, OmniZip (Tao et al., 2025c) represents an early attempt at omni-modal compression, selecting salient audio tokens based on encoder attention and using them to guide video compression. FASTAV (Jung et al., 2026) performs audio-visual token pruning during LLM inference, while EchoingPixels (Gong et al., 2025) takes a more tightly coupled approach by performing global audio–video contextualization before token compression.

# 3. Method

## 3.1. Preliminary

A typical Omni-LLM architecture (Xu et al., 2025a) includes modality-specific encoders, cross-modal projectors, and a generative LLM backbone. Given a video clip $\mathcal{V}$ and synchronized audio $\mathcal{A}$, the encoder-projector pipelines $\Phi_v$ and $\Phi_a$ map each modality into token sequences compatible with the LLM backbone. Specifically,

$$\mathbf{Z}_v = \Phi_v(\mathcal{V}), \quad \mathbf{Z}_a = \Phi_a(\mathcal{A}) \tag{1}$$

Here, $\mathbf{Z}_v$ and $\mathbf{Z}_a$ denote the projected visual and audio tokens after the modality-specific projectors. Specifically, $\mathbf{Z}_v \in \mathbb{R}^{N_v \times D}$ and $\mathbf{Z}_a \in \mathbb{R}^{N_a \times D}$ are the visual and audio token sequences, with $N_v$ and $N_a$ denoting the numbers of visual and audio tokens, and $D$ denoting the LLM hidden dimension.

To maintain temporal alignment, Omni-LLMs group tokens from both modalities into aligned chunks. In our implementation, each multimodal chunk spans 2 seconds, following the local audio-video alignment granularity of Qwen2.5-Omni. Thus, for a video of duration $T$, the number of chunks is $K = \lceil T/2 \rceil$. Let $\mathcal{C}_t$ denote the $t$-th chunk. We define the multimodal block as $\mathcal{C}_t = [\mathbf{Z}_v^{(t)}; \mathbf{Z}_a^{(t)}]$, where $\mathbf{Z}_v^{(t)} \in \mathbb{R}^{n_v \times D}$ and $\mathbf{Z}_a^{(t)} \in \mathbb{R}^{n_a \times D}$ are the visual and audio tokens in the chunk, with $n_v$ and $n_a$ denoting the number of visual and audio tokens per chunk, respectively. The final input sequence $\mathcal{S} = \{\mathcal{C}_1, \ldots, \mathcal{C}_K\}$ is interleaved with textual instructions as the LLM's input.

Each visual sub-sequence $\mathbf{Z}_v^{(t)}$ corresponds to two consecutive frames. Let $n_p \triangleq n_v/2$ denote the number of visual tokens per frame. Let $\mathbf{F}_1^{(t)}, \mathbf{F}_2^{(t)} \in \mathbb{R}^{n_p \times D}$ be the token sequences of the two frames.

## 3.2. OmniSIFT

As illustrated in Figure 3, OmniSIFT operates in two stages: (1) a Spatio-Temporal Video Pruning (STVP) module that removes spatial and temporal redundancy from visual tokens within each chunk, and (2) a Vision-Guided Audio Selector (VGAS) module that selects audio tokens with the refined visual context. Each multimodal chunk $\mathcal{C}_t$ serves as the basic processing unit for OmniSIFT.

Let $\rho_v, \rho_a \in (0, 1]$ denote the visual and audio compression ratios, which represent the proportions of tokens removed from the visual and audio modalities, respectively. The corresponding retention ratios used for token selection are $\alpha_v = 1 - \rho_v, \alpha_a = 1 - \rho_a$.

## 3.3. Spatio-Temporal Video Pruning

The video branch of OmniSIFT aims to remove redundant visual tokens while preserving the spatial and temporal cues needed for downstream omni-modal reasoning. This design is motivated by prior video-token compression studies: VidCom$^2$ (Liu et al., 2025c) shows that many visual tokens are spatially redundant within individual frames, while TimeChat-Online (Yao et al., 2025) highlights temporal redundancy across consecutive frames. The problem we aim to solve is: *how can we retain spatially distinctive regions and temporally changing areas, while discarding redundant patches under a fixed visual compression ratio $\rho_v$?*

To this end, we introduce a Spatio-Temporal Video Pruning (STVP) module that operates at the chunk level and adopts a two-stage pruning strategy: (1) compute spatial saliency scores on $\mathbf{F}_1^{(t)}$ and temporal saliency scores on $\mathbf{F}_2^{(t)}$, and (2) select the top-ranked tokens according to the retention ratio $\alpha_v$.

**Spatial Saliency Estimation.** The first frame encapsulates the static visual layout of the scene. To identify spatially distinctive patches, a frame-level representation $\bar{\mathbf{v}}_1^{(t)}$ is computed via mean pooling to aggregate the global visual context:

$$\bar{\mathbf{v}}_1^{(t)} = \frac{1}{n_p} \sum_{i=1}^{n_p} \mathbf{v}_{1,i}^{(t)}, \tag{2}$$

The spatial saliency of each token $\mathbf{v}_{1,i}^{(t)}$ is subsequently defined as the cosine distance relative to this global mean vector:

$$s_{1,i}^{(t)} = 1 - \frac{\mathbf{v}_{1,i}^{(t)} \cdot \bar{\mathbf{v}}_1^{(t)}}{\|\mathbf{v}_{1,i}^{(t)}\| \, \|\bar{\mathbf{v}}_1^{(t)}\|}. \tag{3}$$

Tokens characterized by higher scores represent patches that exhibit significant divergence from the global frame context and are therefore considered more informative.

**Temporal Saliency Estimation.** The second frame reflects temporal evolution, such as object motion or newly content. Using positional encodings, each token $v_{2,i}^t \in F_2^t$ can be matched to its corresponding patch token $v_{1,i}^t$ in the first frame, enabling the computation of temporal saliency:

$$s_{2,i}^{(t)} = 1 - \frac{\mathbf{v}_{2,i}^{(t)} \cdot \mathbf{v}_{1,i}^{(t)}}{\|\mathbf{v}_{2,i}^{(t)}\| \, \|\mathbf{v}_{1,i}^{(t)}\|}. \tag{4}$$

A higher temporal saliency score indicates a stronger deviation over time, capturing motion dynamics or appearance changes that contribute new information.

**Saliency-guided Token Selection.** Given the spatial and temporal saliency scores, STVP retains the most informative patches under the visual retention ratio $\alpha_v$. Let $\hat{n}_p = \alpha_v n_p$ denote the number of tokens to keep per frame. We select

the top-scoring tokens from each frame:

$$\hat{\mathbf{F}}_1^{(t)} = \text{TopK}(\mathbf{F}_1^{(t)}, \mathbf{s}_1^{(t)}, \hat{n}_p),$$
$$\hat{\mathbf{F}}_2^{(t)} = \text{TopK}(\mathbf{F}_2^{(t)}, \mathbf{s}_2^{(t)}, \hat{n}_p). \qquad (5)$$

The pruned visual sequence is $\hat{\mathbf{Z}}_v^{(t)} = [\hat{\mathbf{F}}_1^{(t)}; \hat{\mathbf{F}}_2^{(t)}]$.

### 3.4. Vision-Guided Audio Selector

Audio streams are typically sampled at high temporal resolutions, which inevitably leads to substantial redundancy. The fundamental challenge is: *given a fixed audio compression ratio $\rho_a$, how can we identify the most salient audio tokens while safely discarding those that are uninformative?*

We rely on the intrinsic modality-asymmetric nature of audio–video data: whether a sound is vital can only be judged when paired with the corresponding visual cues. Motivated by this, we design the Vision-Guided Audio Selector (VGAS) module, which leverages compressed video tokens to guide audio token selection.

Formally, for the $t$-th chunk $\mathcal{C}_t$, VGAS takes the complete audio token sequence $\mathbf{Z}_a^{(t)} \in \mathbb{R}^{n_a \times D}$ and the pruned video token sequence $\hat{\mathbf{Z}}_v^{(t)} \in \mathbb{R}^{\hat{n}_v \times D}$ as inputs:

$$\mathbf{Z}_a^{(t)} \in \mathbb{R}^{n_a \times D}, \quad \hat{\mathbf{Z}}_v^{(t)} \in \mathbb{R}^{\hat{n}_v \times D} \qquad (6)$$

where $\hat{\mathbf{Z}}_v^{(t)}$ is the compressed visual representation generated by the STVP module.

**Vision-Guided Semantic Interaction.** VGAS utilizes a lightweight cross-attention mechanism, where the audio tokens serve as queries $\mathbf{Q}_a$, while the pruned video tokens constitute the keys $\mathbf{K}_v$ and values $\mathbf{V}_v$. Specifically, the attention operation is formulated as:

$$\mathbf{H}_a^{(t)} = \text{Softmax}\left(\frac{\mathbf{Q}_a \mathbf{K}_v^{\top}}{\sqrt{d}}\right) \mathbf{V}_v, \qquad (7)$$

where $d$ denotes the dimension of the attention head. This process produces visually contextualized audio representations $\mathbf{H}_a^{(t)} \in \mathbb{R}^{n_a \times D}$, in which each audio token incorporates visual information to highlight acoustic features that are semantically aligned with the observed scene. To preserve the original audio semantics, VGAS further applies a residual connection:

$$\tilde{\mathbf{H}}_a^{(t)} = \mathbf{H}_a^{(t)} + \mathbf{Z}_a^{(t)}. \qquad (8)$$

**Saliency Scoring and Token Selection.** The residual-enhanced audio representations $\tilde{\mathbf{H}}_a^{(t)}$ are projected through a two-layer MLP followed by a sigmoid activation function to compute a scalar saliency score for each audio token:

$$s_{a,j}^{(t)} = \sigma(\text{MLP}(\tilde{\mathbf{h}}_{a,j}^{(t)})). \qquad (9)$$

These individual scores constitute the saliency sequence $\mathbf{s}_a^{(t)} = \{s_{a,j}^{(t)}\}_{j=1}^{n_a}$. Subsequently, given the audio retention ratio $\alpha_a$, a TopK operator is utilized to select the $\hat{n}_a = \alpha_a n_a$ tokens with the highest scores, resulting in the pruned audio sequence $\hat{\mathbf{Z}}_a^{(t)}$.

**End-to-End Optimization.** To facilitate gradient-based optimization through the non-differentiable TopK selection, VGAS is trained using a straight-through estimator (STE). Specifically, during the forward pass, a binary mask $m_j \in \{0, 1\}$ is generated for each audio token such that $m_j = 1$ if its saliency score $s_{a,j}^{(t)}$ is among the top-$k$ values, and $m_j = 0$ otherwise. Only the tokens selected by this mask are propagated to the LLM backbone. To overcome the zero-gradient issue of discrete selection during the backward pass, we employ an identity surrogate gradient that approximates $\partial m_j / \partial s_{a,j}^{(t)} \approx 1$. This mechanism allows gradients to flow directly to the saliency scores, thereby enabling seamless end-to-end training of the entire architecture.

## 4. Experiment

### 4.1. Experimental Setting

**Model and Data.** Following OmniZip (Tao et al., 2025c), we evaluate OmniSIFT on the Qwen2.5-Omni series (Xu et al., 2025a). To achieve cross-modal alignment for VGAS, we perform fine-tuning on the AVoCaDO SFT dataset (Chen et al., 2025b), which comprises 107K synchronized audio–visual captioning pairs.

**Benchmarks.** We evaluate OmniSIFT on four audio–visual QA benchmarks: VideoMME (with audio) (Fu et al., 2025), DailyOmni (Zhou et al., 2025), WorldSense (Hong et al., 2025), and OmniVideoBench (Li et al., 2025), as well as the video-SALMONN-2 captioning testset (Tang et al., 2025).

**Baselines.** We choose three baselines: (i) **OmniZip** (Tao et al., 2025c), the first compression method designed for Omni-LLMs; (ii) **DyCoke** (Tao et al., 2025b), a video-centric token compression approach whose TTM module we adapt to prune video and audio tokens independently; (iii) **Random Pruning**, which drops video and audio tokens uniformly at random.

**Implementation Details.** The VGAS module uses a lightweight multi-head cross-attention layer with 8 heads and a 512-dimensional hidden size. We fine-tune only the LLM decoder and the VGAS module using a learning rate of $1 \times 10^{-5}$ and a total batch size of 128. For fair comparison, we first fine-tune the Qwen2.5-Omni backbone under the same setting and then apply compression baselines to this model. Additional details are provided in Appendix B.

*Table 1.* **Performance comparison Results**. Results are evaluated on Qwen2.5-Omni-7B and Qwen2.5-Omni-3B across multiple benchmarks, using retained ratios of 35% and 25%. The **best** result among token compression methods for each metric is bolded.

| Method | Retained Ratio (%) | WorldSense (↑) | OmniVideo Bench (↑) | VideoMME (↑) | | | | video-SALMONN-2 (↓) testset | | |
|---|---|---|---|---|---|---|---|---|---|---|
| | | | | Short | Medium | Long | Avg. | Miss | Hal | Total |
| *Qwen2.5-Omni-7B* | | | | | | | | | | |
| Full Tokens | 100 | 49.7 | 35.6 | 78.9 | 66.9 | 57.1 | 67.6 | 29.1 | 19.0 | 48.1 |
| OmniZip | 35 | 48.9 | 35.1 | 77.1 | 67.0 | 56.0 | 66.7 | 34.1 | 20.0 | 54.1 |
| Random | 35 | 48.3 | 33.4 | 77.2 | **68.1** | 56.6 | 67.3 | 33.2 | 19.9 | 53.1 |
| DyCoke | 35 | 48.6 | 34.4 | 78.7 | 68.0 | 56.9 | 67.9 | 32.6 | 20.1 | 52.7 |
| **OmniSIFT** | 35 | **50.0** | **35.6** | **79.0** | 67.9 | **58.0** | **68.3** | **30.7** | **19.8** | **50.5** |
| OmniZip | 25 | 48.1 | 34.1 | 76.4 | 66.1 | 55.3 | 66.0 | 35.8 | 21.4 | 57.2 |
| Random | 25 | 47.1 | 32.6 | 77.0 | 66.1 | 55.1 | 66.1 | 36.2 | 20.7 | 56.9 |
| DyCoke | 25 | 48.1 | 34.1 | 76.4 | 66.2 | 55.0 | 65.9 | 35.3 | **20.0** | 56.3 |
| **OmniSIFT** | 25 | **49.9** | **35.4** | **78.6** | **67.8** | **58.3** | **68.2** | **30.9** | 20.3 | **51.2** |
| *Qwen2.5-Omni-3B* | | | | | | | | | | |
| Full Tokens | 100 | 45.8 | 33.5 | 76.1 | 63.4 | 52.9 | 64.2 | 32.8 | 20.8 | 53.6 |
| OmniZip | 35 | 44.1 | **33.7** | 74.7 | **63.8** | 53.1 | 63.5 | 36.9 | 22.2 | 59.1 |
| Random | 35 | 45.5 | 33.4 | 74.3 | 61.6 | 52.1 | 62.7 | 37.0 | 21.7 | 58.7 |
| DyCoke | 35 | 45.3 | 32.8 | 73.7 | 62.7 | **53.7** | 63.3 | 36.9 | **21.6** | 58.5 |
| **OmniSIFT** | 35 | **45.7** | **33.7** | **76.1** | 62.2 | 52.8 | **63.7** | **35.2** | 21.8 | **56.9** |
| OmniZip | 25 | 43.8 | 32.4 | 72.7 | 61.9 | **52.3** | 62.3 | 39.5 | 22.6 | 62.1 |
| Random | 25 | 43.3 | 33.0 | 74.0 | 61.9 | 50.9 | 62.3 | 39.3 | 22.6 | 62.0 |
| DyCoke | 25 | 44.1 | 33.0 | 73.3 | **62.3** | 51.9 | 62.5 | 40.2 | **21.7** | 61.9 |
| **OmniSIFT** | 25 | **45.8** | **33.1** | **75.0** | 62.0 | 52.1 | **63.0** | **36.4** | 21.9 | **58.3** |

## 4.2. Main Results

**State-of-the-Art Compression Performance.** As shown in Table 1 and Table 2, we evaluate OmniSIFT on five audio-visual benchmarks using Qwen2.5-Omni-7B and 3B under 35% and 25% token retention ratios. Across all settings, OmniSIFT consistently achieves the highest accuracy among compression methods. Notably, the performance of OmniSIFT matches or even exceeds that of the full-token baseline across multiple benchmarks. For instance, while retaining only 35% of the tokens on Qwen2.5-Omni-7B, OmniSIFT achieves a score of 50.0 on WorldSense, surpassing the 49.7 score attained by the full-token model. We attribute it to OmniSIFT's ability to remove redundant audio–visual tokens that may introduce noise, while preserving the key audio-visual cues required for reasoning.

**Fine-Grained Category Results.** Table 2 presents the fine-grained results on DailyOmni for both Qwen2.5-Omni-7B and Qwen2.5-Omni-3B across two retention ratios. In challenging categories that require intricate temporal or cross-modal reasoning, existing token compression methods often suffer from substantial performance degradation. For instance, at a 25% retention ratio with Qwen2.5-Omni-7B, OmniZip achieves only 61.8 on *Event Sequence* and 59.7 on *AV Event Alignment*. These results highlight the limitations of current baselines in capturing temporal dynamics

and cross-modal consistency under aggressive compression. In contrast, OmniSIFT achieves 66.7 and 68.9 in these respective categories, demonstrating its resilience even under extremely constrained token budgets.

**Robustness Across Compression Ratios.** Figure 4 illustrates the performance of OmniSIFT compared to other token compression baselines under various visual and audio compression ratios. As shown in the right panel, as the audio compression ratio $\rho_a$ increases from 0.3 to 0.9, the accuracy of OmniZip drops significantly from over 48.9% to approximately 44.0%. In contrast, OmniSIFT maintains a stable performance above 49.3% across the entire range, exhibiting only minimal degradation even under extreme compression levels.

Overall, these results demonstrate that OmniSIFT achieves the best balance between compression and performance, maintaining reliable audio-visual understanding even when retaining only a small fraction of the original tokens.

## 4.3. Efficiency Analysis

Table 3 presents a comprehensive efficiency comparison on the WorldSense benchmark for both Qwen2.5-Omni-7B and Qwen2.5-Omni-3B, detailing the inference latency and peak GPU memory consumption of OmniSIFT alongside other

*Table 2.* **Performance comparison results on DailyOmni.** Results are evaluated on Qwen2.5-Omni-7B and Qwen2.5-Omni-3B, using retained ratios of 35% and 25%. The **best** result among token compression methods is bolded.

| Method | Retained Ratio (%) | Event Sequence | AV Event Alignment | Inference | Reasoning | Context Understanding | Comparative | Average |
|---|---|---|---|---|---|---|---|---|
| | | | | *Qwen2.5-Omni-7B* | | | | |
| Full Tokens | 100 | 66.7 | 70.6 | 79.2 | 76.6 | 69.9 | 77.1 | 72.2 |
| OmniZip | 35 | 63.7 | 63.0 | 77.3 | 76.6 | 59.1 | 74.8 | 67.7 |
| Random | 35 | 58.5 | 61.8 | 77.9 | 73.7 | 63.2 | 74.0 | 66.3 |
| DyCoke | 35 | 61.4 | 63.9 | 77.9 | 75.4 | 63.7 | 74.8 | 67.9 |
| **OmniSIFT** | 35 | **66.7** | **70.2** | **83.1** | **78.9** | **69.9** | **79.4** | **73.2** |
| OmniZip | 25 | 61.8 | 59.7 | 75.3 | 75.4 | 60.6 | 74.0 | 66.2 |
| Random | 25 | 61.1 | 56.7 | 78.6 | 71.4 | 60.1 | 73.3 | 65.2 |
| DyCoke | 25 | 57.2 | 56.7 | 80.0 | 74.3 | 61.1 | 71.0 | 64.7 |
| **OmniSIFT** | 25 | **66.7** | **68.9** | **82.5** | **77.7** | **71.0** | **76.3** | **72.5** |
| | | | | *Qwen2.5-Omni-3B* | | | | |
| Full Tokens | 100 | 60.1 | 62.2 | 78.6 | 74.9 | 62.2 | 74.8 | 67.0 |
| OmniZip | 35 | **60.5** | 56.7 | 76.6 | 72.0 | 59.6 | **72.5** | 64.7 |
| Random | 35 | 55.9 | 54.2 | 76.0 | 74.3 | 60.1 | 68.7 | 62.9 |
| DyCoke | 35 | 53.9 | 52.5 | **79.2** | **76.0** | 60.1 | **72.5** | 63.2 |
| **OmniSIFT** | 35 | 57.8 | **58.8** | 77.3 | 73.7 | **64.8** | 69.5 | **65.3** |
| OmniZip | 25 | 57.8 | 55.0 | 75.3 | 70.3 | 58.5 | 70.0 | 64.2 |
| Random | 25 | 53.3 | 54.2 | **76.6** | 72.0 | 55.4 | 67.9 | 61.2 |
| DyCoke | 25 | 52.6 | 54.6 | 74.7 | **74.3** | 58.0 | 69.5 | 61.7 |
| **OmniSIFT** | 25 | **58.5** | **59.7** | 75.3 | 73.7 | **60.6** | **70.2** | **64.7** |

token compression baselines. Across both model scales, OmniSIFT achieves substantial reductions in computational overhead compared to the full-token model. Specifically, for the 7B variant, OmniSIFT reduces total inference time by over 40% and lowers peak memory usage by more than 4.6 GB, with consistent improvements observed for the 3B variant. Notably, despite the inclusion of a learned cross-modal module, the end-to-end latency and peak memory requirements of OmniSIFT remain on par with training-free approaches such as OmniZip and DyCoke, demonstrating its high operational efficiency.

## 4.4. Ablation Study

We conduct ablation studies to examine two primary aspects of OmniSIFT: the individual contributions of the video and audio compression modules, and the impact of the asymmetric token compression paradigm. For consistency, all ablation experiments are performed using the Qwen2.5-Omni-7B model as the base architecture.

**Structural Ablation: STVP and VGAS.** Figure 5 illustrates the performance impact of the STVP and VGAS modules within OmniSIFT. For the STVP module, we assess the individual contributions of its spatial and temporal components. In the "w/o Spatial Component" variant, all visual

tokens are selected using temporal saliency only, while in the "w/o Temporal Component" variant, all visual tokens are selected using spatial saliency only. The removal of either component results in a noticeable reduction in accuracy on both DailyOmni and WorldSense, underscoring their complementary roles. Regarding the VGAS module, we compare OmniSIFT against an "Audio-Only Selector" baseline, where audio token selection relies exclusively on intra-audio dependencies. In this configuration, the cross-modal attention mechanism in VGAS is replaced by an audio self-attention module within each chunk. This modification leads to significant accuracy declines of 3.9% and 2.9% on DailyOmni and WorldSense, respectively. These results demonstrate that the importance of audio tokens is highly context-dependent and necessitates visual guidance for accurate assessment. Additional ablation results for the VGAS module are detailed in Appendix D.4.

**Token Compression Paradigm Ablation.** Table 4 compares our modality-asymmetric paradigm, which employs vision-guided audio selection, with a modality-symmetric paradigm that utilizes audio-guided video pruning. Both paradigms are evaluated across three different retention ratios on DailyOmni and WorldSense. To implement the symmetric baseline, we fine-tune the Qwen2.5-Omni-7B backbone following the pruning methodology of OmniZip.

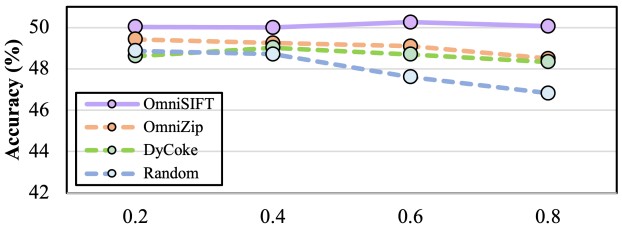 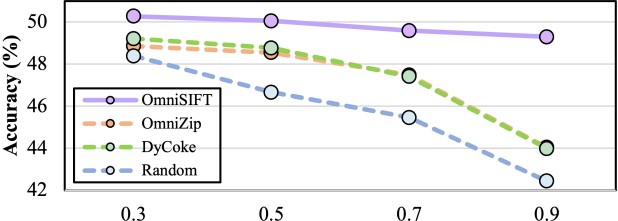

*Figure 4.* **Ablation results for video and audio compression ratios**, evaluated on the Qwen2.5-Omni-7B model using the WorldSense benchmark. **Left**: Varying the video compression ratio $\rho_v$ with audio compression ratio $\rho_a = 0.5$; **Right**: Varying the audio compression ratio $\rho_a$ with video compression ratio $\rho_v = 0.8$.

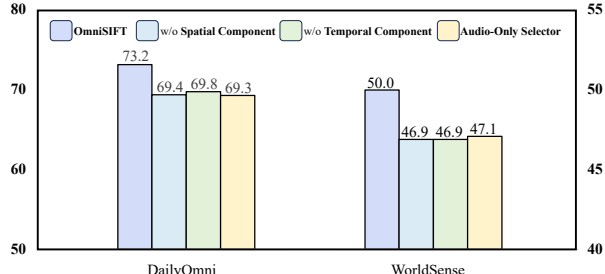

*Figure 5.* **Ablation results for OmniSIFT's architecture**. **w/o Spatial Component**: all visual tokens are selected using temporal saliency only. **w/o Temporal Component**: all visual tokens are selected based on spatial saliency only. **Audio-Only Selector**: audio tokens are selected solely based on intra-audio self-attention without any visual guidance.

Across all retention ratios, OmniSIFT consistently outperforms the symmetric variant, with the performance gap becoming more pronounced as the retention ratio decreases. These results demonstrate that the asymmetric strategy of OmniSIFT effectively preserves more salient tokens by explicitly modeling the cross-modal dependencies between visual and audio modalities.

### 4.5. Case Study

Figure 6 presents a case study comparing OmniSIFT with OmniZip on OmniVideoBench, illustrating a key limitation of modality-symmetric compression methods: assume audio and video signals at the same time carry comparable importance. In this example, when the score changes, the audio signal receives a low saliency score and allocates a small compression budget to video; as a result, the scoreboard patches are pruned, yielding an incorrect answer. In contrast, OmniSIFT adopts a modality-asymmetric compression that preserves the salient video patches and contextually informative audio cues necessary for correct reasoning.

### 5. Conclusion

In this work, we introduce OmniSIFT, a modality-asymmetric token compression framework for Omni-LLMs.

*Table 3.* **Efficiency comparison results**. Results are evaluated on Qwen2.5-Omni-7B and Qwen2.5-Omni-3B using the WorldSense benchmark, reporting peak GPU memory usage, inference latency. The **best** result among token compression methods for each metric is in bold, the second best result is underlined.

| Method | Retained Ratio (%) | GPU Mem (GB)↓ | Total Time (s)↓ | Prefill Lat. (s)↓ | E2E Lat. (s)↓ | Acc (%)↑ |
|---|---|---|---|---|---|---|
| **Qwen2.5-Omni-7B** | | | | | | |
| Full Tokens | 100 | 27.59 | 15097.1 | 4.76 | 4.94 | 49.7 |
| OmniZip | 35 | 22.92 | 8886.4 | 2.80 | 2.89 | 48.9 |
| DyCoke | 35 | 23.09 | **8718.3** | **2.75** | **2.85** | 47.3 |
| **OmniSIFT** | 35 | **22.91** | 8756.0 | 2.76 | 2.86 | **50.0** |
| **Qwen2.5-Omni-3B** | | | | | | |
| Full Tokens | 100 | 18.91 | 11399.4 | 3.59 | 3.79 | 45.8 |
| OmniZip | 35 | **14.75** | 7750.4 | 2.44 | 2.59 | 44.1 |
| DyCoke | 35 | 14.92 | **7578.8** | **2.39** | **2.53** | 43.9 |
| **OmniSIFT** | 35 | 14.79 | 7596.3 | **2.39** | **2.53** | **45.7** |

Inspired by the asymmetric nature of human audio–video perception, OmniSIFT first decouples spatial and temporal redundancy in video tokens to obtain compact visual cues, and then uses these cues to guide audio token selection. Experiments on five audio–visual benchmarks show that OmniSIFT consistently outperforms existing compression baselines and, in several settings, even exceeds the performance of full-token models. It also delivers substantial gains in inference speed and memory usage. Overall, OmniSIFT provides an effective and efficient approach for reducing token counts in Omni-LLMs while preserving the key audio–visual information required for downstream tasks.

### 6. Limitations.

Although OmniSIFT achieves strong tradeoffs between performance and efficiency under aggressive compression, precise alignment between audio and visual signals remain challenging, since such tasks often depend on highly localized evidence that is sensitive to token pruning. Moreover, STVP is a lightweight chunk-level module and currently

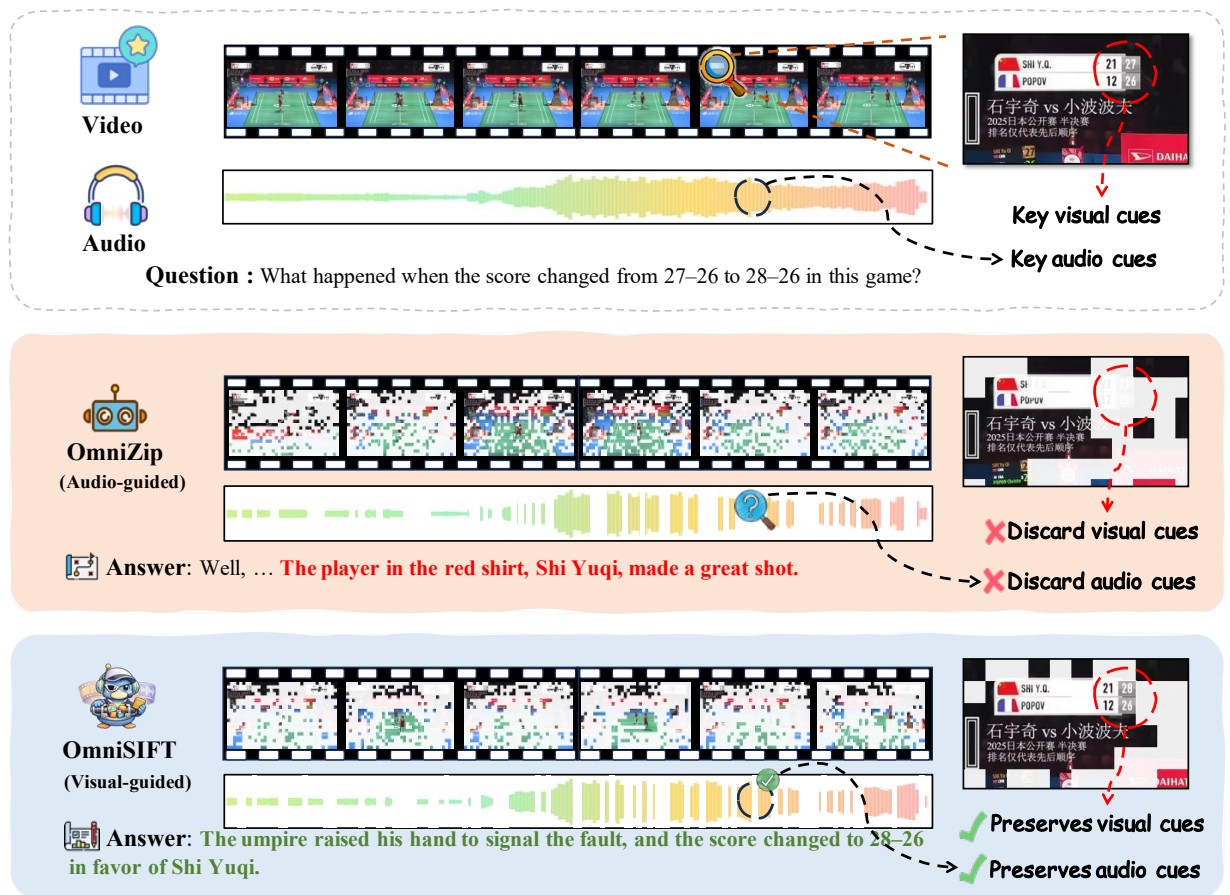

*Figure 6.* **Visualization of token compression methods for Omni-LLMs.** White blocks denote discarded video and audio tokens. The vertical amplitude of the waveform reflects the audio information density. As illustrated, OmniZip prunes critical visual features and audio cues, leading to an erroneous interpretation of the score change. In contrast, OmniSIFT preserves both the salient visual dynamics and the informative audio segments required for accurate event reasoning.

*Table 4.* **Ablation results for compression paradigm.** We compare our video-guided audio compression with an OmniZip-style trained compression method. All experiments use the Qwen2.5-Omni-7B backbone and evaluate three retained ratios on Daily-Omni and WorldSense. The **best** results are bolded.

| Benchmark | Retained Ratio (%) | Daily-Omni | World-Sense |
|---|---|---|---|
| OmniZip-Trained | 35 | 70.5 | 49.7 |
| OmniSIFT | 35 | **73.2** | **50.0** |
| OmniZip-Trained | 30 | 69.3 | 49.3 |
| OmniSIFT | 30 | **72.8** | **50.0** |
| OmniZip-Trained | 25 | 68.8 | 48.7 |
| OmniSIFT | 25 | **72.5** | **49.9** |

models temporal redundancy only within each multimodal chunk, leaving temporal redundancy across chunks and long-range audio-visual structure unexplored. Finally, OmniSIFT uses a fixed and query-agnostic token budget to support broad omni-modal tasks beyond single-turn QA. Future work may explore adaptive budgeting for videos with non-uniform information density and query-guided pruning for task-specific compression, while carefully preserving multi-turn context and audio-triggered visual evidence.

## Acknowledgements

This work was supported by the Beijing Natural Science Foundation (L252033) and the National Natural Science Foundation of China (92570204, 62576339).

## Impact Statement

OmniSIFT improves the efficiency of Omni-modal LLMs by reducing redundant tokens while preserving or enhancing performance, enabling wider deployment in resource-constrained or real-time settings. By encouraging semantically meaningful, cross-modal representations, it benefits applications such as audio-visual QA and video captioning.

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

# A. Expanded Benchmark Details

To rigorously evaluate OmniSIFT across a broad spectrum of audio-visual understanding tasks, we select five representative benchmarks that encompass: (i) long-horizon temporal reasoning, (ii) cross-modal alignment and fusion, (iii) fine-grained multi-dimensional comprehension, and (iv) generative captioning. Our selection process is governed by two key criteria. The first is *capability coverage*, which focuses on competencies essential for practical omni-modal assistants, such as temporal integration and causal reasoning. The second is *protocol availability*, which prioritizes benchmarks with standardized evaluation protocols to ensure the reproducibility of results. Table 5 summarizes the benchmarks, dataset scales, and metrics employed.

*Table 5.* Evaluation benchmarks used in this work.

| Benchmark | #Videos | #QA | #Caps | Metric |
|---|---|---|---|---|
| DailyOmni | 684 | 1,197 | – | Acc. (overall & by QA type) |
| Video-MME | 900 | 2,700 | – | Acc. |
| WorldSense | 1,662 | 3,172 | – | Acc. |
| OmniVideoBench | 628 | 1,000 | – | Acc. |
| video-SALMONN-2 | 483 | – | 483 | GPT-judge (Comp., Hall.) |

"–" indicates not applicable (QA-only or caption-only evaluation).

# B. Expanded Implementation Details

**Input Configuration and Preprocessing** For both training and inference, video inputs are uniformly sampled at a rate of 2 frames per second (FPS), with the total frame count restricted to a maximum of 256. The spatial resolution for each individual frame is configured at a maximum of $320 \times 28 \times 28$ pixels.

**Configuration of Visual and Audio Compression Ratios** Table 6 details the specific visual ($\rho_v$) and audio ($\rho_a$) compression ratios selected for various methods across different total retention levels. For the 35% total retention setting, the compression ratios for each method are initialized based on the protocols defined in OmniZip (Tao et al., 2025c). Given the architectural differences between token compression methods, we dynamically calibrate these values to ensure that the actual quantity of retained audio and video tokens remains consistent across all baselines. For the 25% retention setting, the optimal balance between visual and audio compression is determined through empirical evaluation, while maintaining the same principle of parity in the final token budget across different methods.

**Evaluation Prompts** The input prompts utilized for evaluating QA benchmarks, such as VideoMME, DailyOmni, WorldSense, and OmniVideoBench, are formatted in accordance with the protocol established by (Fu et al., 2025), as illustrated in Figure 7. For the Video-SALMONN-2 benchmark, the input prompt used to elicit detailed descriptions

*Table 6.* $\rho_v$ (video) and $\rho_a$ (audio) for different retrained ratios.

| Methods | 35% | | 25% | |
|---|---|---|---|---|
| | $\rho_a$ | $\rho_v$ | $\rho_a$ | $\rho_v$ |
| OmniZip | 0.4 | 0.7 | 0.6 | 0.98 |
| DyCoke | 0.4 | 0.9 | 0.6 | 0.99 |
| Random | 0.4 | 0.67 | 0.5 | 0.77 |
| OmniSIFT | 0.4 | 0.67 | 0.5 | 0.77 |

also follows the original methodology (Tang et al., 2025), as shown in Figure 8. To assess the quality of these generated captions, we adopt an LLM-as-a-judge framework where GPT-4.1 serves as the evaluator. The specific judgment prompt utilized by the evaluator model is presented in Figure 9.

# C. Computing Cost Evaluation

The computational overhead of OmniSIFT primarily consists of the parameter requirements within the VGAS module and the operational complexity of the STVP module. We analyze these costs specifically for the Qwen2.5-Omni-7B backbone, where $d_{model} = 3584$.

### C.1. Parameter Efficiency

The VGAS module is designed to be highly lightweight, ensuring minimal impact on the overall memory footprint. For the Qwen2.5-Omni-7B configuration ($d_{model} = 3584$), the module utilizes projections to an internal dimension of 512, followed by a single-layer cross-attention mechanism and a compact MLP-based score head. The cumulative parameter count for these components is approximately 4.85M. This additional overhead represents less than 0.1% of the total parameters in the 7B-class LLM backbone, demonstrating the extreme parameter efficiency of OmniSIFT.

*Table 7.* Efficiency comparison in theoretical FLOPs in evaluation on the WorldSense with Qwen2.5-Omni-7B.

| Method | Retained Ratio | Selector FLOPs (T) | LLM FLOPs(T) | Total FLOPs (T) |
|---|---|---|---|---|
| Full Tokens | 100% | / | 555.74 | 555.74 |
| OmniSIFT | 35% | 0.06 | 292.10 | 292.16 |
| OmniSIFT | 25% | 0.04 | 250.79 | 250.83 |

### C.2. Computational Complexity

The computational complexity of the compression modules is significantly lower than the self-attention mechanism of the LLM backbone. Specifically, STVP operations scale linearly with the sequence length, while the VGAS module performs efficient cross-attention within localized chunks using reduced dimensions. Compared to the quadratic overhead of the backbone, the additional FLOPs introduced by

---

**Prompt for QA Evaluation**

Select the best answer to the following multiple-choice question based on the video. Respond with only the letter (A, B, C, or D) of the correct option.
What visual elements were displayed immediately after Dr. Rajani's 'BOTOX WITHOUT THE BOTOX' video concluded?
A. Still product bottle $\rightarrow$ Price text overlay
B. Facial treatment demonstration $\rightarrow$ Presenter holding product while explaining
C. Presenter's torso shot $\rightarrow$ Secondary screen activation
D. Bookshelf backdrop $\rightarrow$ Close-up of string lights
The best answer is:

---

*Figure 7.* Input prompt template utilized for question-answering (QA) benchmarks, following the protocol of VideoMME (**?**).

---

**Prompt for Video Description**

Please provide a thorough description of all the content in the video, including every detail. As you describe, ensure that you also cover as much information from the audio as possible, and be mindful of the synchronization between the audio and video as you do so.

---

*Figure 8.* Inference prompt employed to elicit detailed descriptions for evaluation on the Video-SALMONN-2 benchmark.

---

these modules are negligible. Table 7 provides an empirical comparison of the FLOPs required by the full-token model versus OmniSIFT at 25% and 35% retention ratios, further demonstrating the computational efficiency of the proposed framework. Specifically, at a 25% retention ratio, OmniSIFT requires only 250.83T FLOPs, a reduction of over 50% compared to the 555.74T required by the full-token baseline.

# D. More Experimental Results

## D.1. Visualization of Attention Sparsity

To investigate the internal mechanisms of multi-modal understanding, we visualize the attention score distributions of the Qwen2.5-Omni-7B backbone at Layer 15 and Layer 27. As illustrated in Figure 10, a high degree of attention sparsity is observed across both layers, with the majority of video and audio tokens receiving near-zero attention scores; this indicates that the original dense representation contains substantial redundant information that does not influence the final output generation. This sparsity pattern becomes even more pronounced in the deeper layers, as evidenced by the noticeably lower attention scores in Layer 27 compared to Layer 15, suggesting that the model progressively filters out irrelevant spatio-temporal details to prioritize high-level semantics. Such empirical observations provide a strong motivation for OmniSIFT, which significantly reduces the computational load without compromising the representative capacity of the model.

## D.2. Efficiency Gains across Video Lengths

To investigate the scalability of OmniSIFT, we analyze its performance in terms of both computational efficiency and memory consumption as video duration increases from 0s to 120s. As illustrated in Figure 11, OmniSIFT effectively transforms the scaling behavior of the system by preventing the prohibitive resource growth. While the end-to-end (E2E) latency of the full-token baseline escalates significantly due to the quadratic complexity of self-attention, OmniSIFT maintains a substantially more sustainable growth trajectory, achieving a latency reduction of over 60% for videos exceeding 60 seconds. Similarly, the proposed framework exhibits superior memory scalability; although the peak GPU memory consumption of the baseline model increases rapidly with extended temporal contexts, the growth rate for OmniSIFT remains modest, achieving a reduction of approximately 28% when the video duration reaches 120s. These empirical results demonstrate that OmniSIFT is not merely a localized optimization but a necessary prerequisite for scaling Omni-LLMs to handle extended video sequences within restricted computational budgets.

## D.3. Ablation on Selector Depth

To verify whether the complexity of the VGAS module impacts pruning quality, we conduct a comparative study between the default single-layer ($N = 1$) design and a 3-layer ($N = 3$) variant. As summarized in Table 8, increasing the depth of the cross-modal interaction does not yield performance gains; instead, it leads to a slight degradation in

---

**Prompts used to evaluate captions in video-SALMONN-2**

**Task:** A good video description should capture the detailed events in the video. The task is to judge whether a given description is good or not. The model is provided with a list of base events and a candidate description, and must determine which base events are covered by the description.

**Instruction:** Besides correctly described events, the description may contain missed events, incorrect events, or hallucinated events.

- **Missed Event:** An event in the base set whose main action, participants, and context are absent from the description.

- **Incorrect Event:** An event in the base set that is mentioned but described with significant factual errors.

- **Hallucination Event:** An event mentioned in the description but not included in the base set and not a plausible inference from it.

The model must also enumerate these events. Incorrect and hallucination events must not overlap.

**Input Format:**

- There are {event_num} base events given as a Python list: ["xxx", ...].

- A video description to be evaluated is provided.

**Output Format (strict):** {"Missed": x, "Incorrect": x, "Hallucination": x, "Missed Event": [...], "Incorrect Event": [...], "Hallucination Event": [...]}

**Events in the Video:** {events_in_video}

**Video Description to be Rated:** {cap_to_be_rated}

Given the base events and the candidate description, count missed, incorrect, and hallucinated events and list them out.

---

*Figure 9.* Prompt for caption evaluation on video-SALMONN-2 test set.

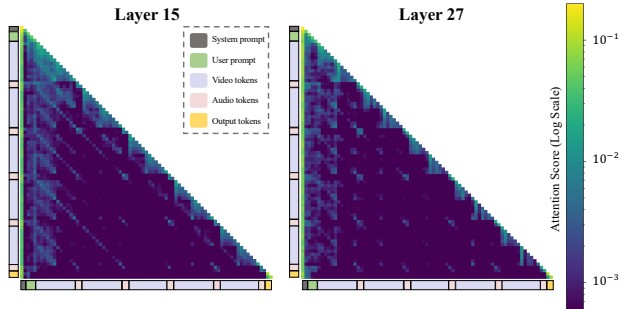

*Figure 10.* Attention score distribution maps for layers 15 and 27 of the LLM decoder in the Qwen2.5-Omni-7B model.

*Table 8.* Performance comparison of various selector depths at a 35% retention ratio across representative benchmarks. All results are obtained using the Qwen2.5-Omni-7B model. The **best** result for each metric is bold.

| Configuration | Video-MME (%) | Daily-Omni (%) | World-Sense (%) | GPU Mem (GB) ↓ |
|---|---|---|---|---|
| 1-Layer (Ours) | **68.3** | **73.2** | **50.0** | **22.62** |
| 3-Layer Variant | 67.2 | 72.3 | 49.0 | 22.67 |

accuracy across all evaluated benchmarks. Specifically, at a 35% retention ratio, the 3-layer variant achieves scores of 67.2, 72.3, and 49.0 on VideoMME, DailyOmni, and WorldSense, respectively, which are lower than the 68.3, 73.2, and 50.0 obtained by the single-layer configuration. Furthermore, the 3-layer design marginally increases the peak GPU memory consumption from 22.62 GB to 22.67 GB. These results suggest that a shallow architecture is sufficient to capture the cross-modal correlations necessary for token selection, and increasing the depth may introduce unnecessary complexity that hinders the identification of salient audio tokens. Consequently, the single-layer configuration provides the optimal balance between computational efficiency and pruning effectiveness.

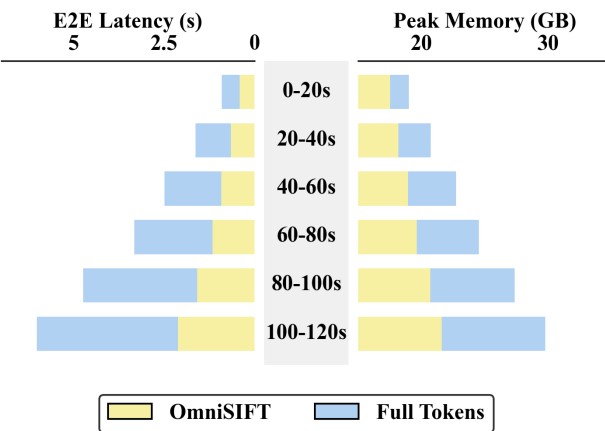

*Figure 11.* Comparison of peak GPU memory and end-to-end latency between OmniSIFT and full-token baseline using Qwen2.5-Omni-7B on WorldSense videos of varying durations.

## D.4. Extended Ablation Results

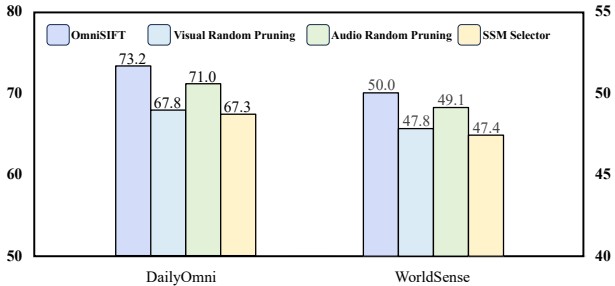

*Figure 12.* Results of extended ablation experiments on the architecture of OmniSIFT, conducted using the Qwen2.5-Omni-7B model at a 35% retention ratio. **Visual Random Pruning**: replacing the STVP module with random selection for video tokens; **Audio Random Pruning**: replacing the VGAS module with random selection for audio tokens; **SSM Selector**: utilizing a State Space Model as the selector for audio tokens.

In addition to the experiments in Section 4.4, to further validate the architectural components of OmniSIFT, we conduct extended ablation studies using the Qwen2.5-Omni-7B backbone at a 35% retention ratio. In these experiments, we compare the proposed modules against three alternatives: (i) replacing the STVP module with visual random pruning, (ii) substituting the VGAS module with audio random pruning, and (iii) employing a State Space Model (SSM) as the audio token selector. All variants are trained using identical datasets and optimization settings to ensure a fair comparison. As illustrated in Figure 12, OmniSIFT achieves superior performance on both DailyOmni (73.2) and World-Sense (50.0). Notably, the replacement of STVP with visual random pruning leads to a more severe decline in accuracy (67.8 on DailyOmni and 47.8 on WorldSense) compared to audio random pruning (71.0 and 49.1, respectively). This observation underscores the modality-asymmetric nature of

audio-visual inputs, suggesting that the loss of critical visual tokens is more detrimental and harder for the model to recover than the removal of audio tokens. Furthermore, the SSM-based selector significantly underperforms the VGAS module, yielding the lowest scores (67.3 and 47.4). These results confirm that both the STVP and VGAS modules are essential for effectively capturing cross-modal dependencies and preserving salient information.

## D.5. Additional Rebuttal Experiments

We include additional experiments conducted during the rebuttal period to further validate OmniSIFT from four aspects: comparison with a recent audio-visual token pruning baseline, transfer to another omni-modal backbone, adaptive video-token budget allocation, and alternative designs for the vision-guided audio selector.

**Comparison with FASTAV.** We reproduce FASTAV (Jung et al., 2026) and evaluate it on DailyOmni and WorldSense. Following the original paper, we use its two-stage pruning strategy: 50% global pruning at a middle decoder layer, followed by 20% fine pruning in subsequent layers. As shown in Table 9, OmniSIFT consistently outperforms FASTAV under a similar token budget.

*Table 9.* Comparison with FASTAV on DailyOmni and WorldSense. Results are evaluated under a similar token budget.

| Method | DailyOmni (↑) | WorldSense (↑) |
|---|---|---|
| OmniSIFT (35% tokens) | **73.2** | **50.0** |
| FASTAV | 59.4 | 43.3 |

**Transfer to Qwen3-Omni.** To evaluate cross-backbone generalization, we apply OmniSIFT to Qwen3-Omni (Xu et al., 2025b). Table 10 shows that OmniSIFT transfers to this stronger omni-modal backbone with only minor performance degradation, while reducing prefilling latency and GPU memory usage.

**Adaptive Budget Allocation.** We compare the fixed chunk-level video token budget used in OmniSIFT with an adaptive allocation strategy following VidCom$^2$ (Liu et al., 2025c). Both variants are evaluated under the same 35% token retention budget. As shown in Table 11, the adaptive strategy does not provide clear performance gains and also increases prefill latency due to the additional budget estimation step. These results support our fixed allocation as a practical design choice that favors efficiency and stability.

**Alternative VGAS Designs.** We further compare VGAS against two audio-compression variants trained under the same setting. Direct_attention removes the score

*Table 10.* Transfer results on Qwen3-Omni. Latency is measured in seconds and GPU memory in GB.

| Backbone / Method | DailyOmni (↑) | WorldSense (↑) | DailyOmni Prefill / GPU Mem (↓) | WorldSense Prefill / GPU Mem (↓) |
|---|---|---|---|---|
| Qwen3-Omni (Full Tokens) | **70.8** | **50.2** | 2.22 / 63.12 | 2.97 / 72.18 |
| Qwen3-Omni + OmniSIFT (LoRA, 35%) | 70.5 | 48.8 | **2.16 / 60.85** | **2.48 / 68.56** |

*Table 11.* Comparison between fixed and adaptive budget allocation under a 35% token retention budget. Prefill latency is measured in seconds.

| Method | DailyOmni (↑) | WorldSense (↑) | DailyOmni Prefill (↓) | WorldSense Prefill (↓) |
|---|---|---|---|---|
| OmniSIFT | **73.2** | **50.0** | **1.02** | **2.76** |
| Adaptive | 71.9 | 49.3 | 1.40 | 3.02 |

head in VGAS and directly uses audio-visual attention for audio token pruning. `Concate_av` removes the cross-attention module and instead concatenates the audio tokens with the video tokens within the same chunk before applying a score head. As summarized in Table 12, the proposed VGAS design achieves the best overall performance under the same 35% token retention budget.

*Table 12.* Comparison of alternative audio token compression designs at a 35% token retention budget.

| Method | DailyOmni (↑) | WorldSense (↑) |
|---|---|---|
| OmniSIFT | **73.2** | **50.0** |
| Direct_attention | 72.4 | 49.7 |
| Concate_av | 71.8 | 49.4 |

