# OpenReview forum: "OmniSIFT: Modality-Asymmetric Token Compression for Efficient Omni-modal Large Language Models"
_ICML.cc/2026/Conference — ICML 2026 regular_

### Official Review · Reviewer_9TNa · 2026-02-23

**Soundness:** 2
**Presentation:** 3
**Significance:** 2
**Originality:** 3
**Overall Recommendation:** 4
**Confidence:** 3

**Summary:**

The authors propose OmniSIFT, a modality-asymmetric token compression framework designed for Omni-modal LLMs. To accelerate inference on video-audio understanding tasks while preserving model capabilities, OmniSIFT operates sequentially through two modules: Spatio-Temporal Video Pruning (STVP) and a Vision-Guided Audio Selector (VGAS). STVP reduces visual redundancies in frame level, while VGAS employs a lightweight, trainable cross-attention mechanism to identify and retain audio tokens most relevant to the preserved visual context. By decoupling the token selection from the LLM's internal attention scores, OmniSIFT is readily compatible with advanced attention mechanisms like FlashAttention. Empirical results demonstrate the effectiveness of this modality-asymmetric pruning paradigm.

**Compliance With Llm Reviewing Policy:**

Affirmed.

**Final Justification:**

Most of my concerns have been solved; however, I think this paper still lacks outstanding insight in omnimodal token pruning, which is more like a combination of existing techniques. Thus, I suggest for weak acceptance recommendation.

**Key Questions For Authors:**

1. Prior research clearly reveals the importance of query-awareness in visual token pruning. However, the first stage (STVP) in OmniSIFT appears entirely query-agnostic. Given that the authors successfully employ a trainable cross-modal attention mechanism for video-guided audio selection, why not adopt a similar learnable, query-guided approach for video token pruning? The decision to apply a parameter-free method for the video modality while relying on learnable attention scores for the audio modality seems structurally inconsistent. The authors should clarify whether this architectural asymmetry provides a fundamental methodological advantage, or if it is primarily an engineering compromise. The underlying motivation for this specific design choice needs to be articulated more convincingly.
2. STVP operates at the chunk level. Eq. (4) evaluates intra-chunk temporal redundancy (between $\mathbf{F}_1^{(t)}$ and $\mathbf{F}_2^{(t)}$). How does OmniSIFT account for inter-chunk temporal redundancy (i.e., between $\mathbf{F}_1^{(t)}$ and $\mathbf{F}_1^{(t+1)}$)?
3. I have concerns regarding the frame-level budget allocation strategy within the STVP module. According to the preliminaries in Section 3.1, the number of visual tokens $n_p$ is identical across all sub-sequences $\mathbf{Z}_v$. Consequently, given a visual retention ratio $\alpha_v$, the retained token count $\hat{n}_p=\alpha_vn_p$ remains strictly static for every chunk $\mathcal{C}$. Furthermore, the budget explicitly allocated to individual frames ($\mathbf{F}_1^{(t)}$ and $\mathbf{F}_2^{(t)}$) is invariably fixed at $\hat{n}_p$. This implies a uniform visual token budget across all frames throughout the entire video.
This design is highly counter-intuitive as it fundamentally ignores the temporal fluctuation of information entropy inherent in video data. Naturally, highly informative frames should dynamically receive a larger token budget, whereas less informative frames (e.g., static scenes or plain transitions) should be allocated significantly fewer tokens or even be skipped entirely. Implementing a delicate, dynamic temporal compression strategy is critical for video processing, as it constitutes the fundamental methodological distinction between video token compression and static image token pruning. The authors should provide robust theoretical or empirical justification for this static allocation strategy, or explicitly address this as a limitation.
4. While Table 3 shows promising end-to-end latency reductions overall, maintaining a strict sequential computation pipeline (Video pruning then Audio pruning) intrinsically prevents parallel processing of modalities. In more extreme scenarios (e.g., massive context windows or streaming inputs), does this sequential barrier introduce a performance-latency trade-off?

**Limitations:**

See Minor weakness 2

**Strengths And Weaknesses:**

**Strengths**

1. The introduction of a modality-asymmetric paradigm for omni-modal token pruning is insightful. Explicitly establishing a causal dependency (video driving audio selection) offers a fresh perspective for future multimodal compression research.
2. By avoiding reliance on the LLM's internal attention scores, the externalized VGAS module allows OmniSIFT to be seamlessly integrated with efficient mechanisms like FlashAttention. This design choice is convincingly validated in the efficiency analysis.
3. Extensive evaluations across multiple benchmarks (e.g., Video-MME, WorldSense) show consistent improvements over strong baselines like OmniZip and DyCoke. The method impressively maintains high task performance even under aggressive token retention budgets (e.g., 25%).

**Major Weaknesses**

1. As presented in Eq.~(5), STVP adopts a fixed per-frame retention budget ($\hat{n}_p=\alpha_v n_p$) within each chunk. While this design is simple and efficient, it may be suboptimal under non-uniform temporal information density (e.g., static/transition frames versus key informative frames), where enforcing the same token budget for all frames can waste capacity. The paper would be significantly strengthened by an ablation comparing this fixed per-frame allocation against an adaptive chunk-level budget allocation under the same total visual token budget (e.g., $\hat{n}_1+\hat{n}_2=\alpha_v n_v$), to verify whether the static split is indeed the best design choice.
2. In the Spatial Saliency Estimation, STVP relies on cosine distance incorporated with the frame-level anchor to maximize visual token diversity. This strategy has been widely explored in prior visual token pruning works, e.g.,
    - [1] Alvar S R, Singh G, Akbari M, et al. Divprune: Diversity-based visual token pruning for large multimodal models. CVPR'25.
    - [2] Wen Z, Gao Y, Wang S, et al. Stop Looking for Important Tokens in Multimodal Language Models: Duplication Matters More. EMNLP'25.
    - [3] Li Y, Zhan H, Chen T, et al. Why 1+ 1< 1 in Visual Token Pruning: Beyond Naive Integration via Multi-Objective Balanced Covering. NeurIPS'25.
    - [4] Dong S, Hu J, Zhang M, et al. MMTok: Multimodal Coverage Maximization for Efficient Inference of VLMs. ICLR'26.

    The authors need to explicitly discuss the differences between STVP and these existing spatial pruning techniques. Furthermore, it is unclear why a query-guided selection approach was not adopted (e.g., retaining tokens most relevant to the user's prompt before maximizing diversity).
3. The evaluations are exclusively conducted on the Qwen2.5-Omni series. While STVP is parameter-free, VGAS requires fine-tuning to evaluate cross-modal dependencies. Could the optimized VGAS be overfitting to the modality-specific encoder representations of Qwen2.5-Omni? The authors should discuss the cross-architecture generalizability of OmniSIFT and ideally provide empirical validation on at least one other Omni-modal LLM architecture on representative benchmarks.

**Minor Weaknesses**

1. The paper requires minor structural polish. For instance, Figure 1 appears in the text after Figure 2, disrupting the logical reading flow. Swapping their positions is recommended.  Besides, the cross-referencing of figures in the main text is inconsistent, e.g., Fig. 2 in line 52, Figure 1 in line 76.
2. This paper lacks a dedicated discussion on failure cases and limitations, which is crucial for understanding the bounds of the proposed method in real-world applications.

---

> ### Author Rebuttal · Authors · 2026-03-30
>
> **Response to Reviewer 9TNa**
>
> Thank you for your recognition of our work and for your thoughtful review. Below, we provide point-by-point responses to your comments.
>
> **W1.** The paper lacks a comparison between fixed and adaptive frame-level budget allocation under the same total token budget.
>
> **A1.** We implement an adaptive budget allocation strategy following VidCom$^2$. Under the 35% token retention budget, the results are as follows:
>
> | Method | DailyOmni | WorldSense | DailyOmni Prefill Latency | WorldSense Prefill Latency |
> |:------:|:---------:|:----------:|:-------------------------:|:--------------------------:|
> | OmniSIFT | 73.2 | 50.0 | 1.02 | 2.76 |
> | Adaptive | 71.9 | 49.3 | 1.40 | 3.02 |
>
> These results show that the adaptive design does not provide clear performance gains, while also increasing prefill latency. Intuitively, adaptive allocation requires additional heuristic estimation to determine budgets, which increases computation. Therefore, the current static allocation is a practical design choice that favors efficiency and stability.
>
> **W2(a).** The distinction between STVP and prior spatial pruning methods is not sufficiently clear.
>
> **A2(a).** Please see our response in **A2 to Reviewer w67Y**.
>
> **W2(b).** It is unclear why STVP is not query-guided.
>
> **A2(b).** We intentionally do not adopt a query-guided design for STVP. One reason is scope: query-guided compression is more naturally suited to query-conditioned QA, while we hope OmniSIFT can support broader omni-modal tasks such as omni-captioning (e.g., the video-SALMONN-2 testset). Another reason is efficiency: query-guided designs are typically either Q-Former-like, which require substantial training data, or attention-based (e.g., SparseVLM), which are less compatible with FlashAttention and thus weaken the efficiency advantage of token compression.
>
> **W3.** The authors should discuss the cross-architecture generalizability of OmniSIFT.
>
> **A3.** OmniSIFT is not tied to a Qwen2.5-Omni-specific internal pattern. STVP is a video-only redundancy reduction module, and VGAS operates on the projected multimodal token sequence. Therefore, OmniSIFT is expected to be applicable to Omni-LLMs that follow the common design pattern of first obtaining modality-specific embeddings and then concatenating multimodal tokens before the LLM backbone.
>
> To further validate this point, we additionally evaluate OmniSIFT on Qwen3-Omni (see our response in **A2(a) to Reviewer 21LA**), where it achieves only minor performance drop while still reducing prefill latency and GPU memory usage. We will include these results in the revised version.
>
> **W4.** The paper requires minor structural polish.
>
> **A4.** We will carefully check and revise the relevant issues in the revised version.
>
> **W5.** This paper lacks a dedicated discussion on failure cases and limitations.
>
> **A5.** We conduct additional case-level failure analysis on DailyOmni (see our response in **A1(b) to Reviewer 4RNK**), and we will incorporate these analyses together with a dedicated limitations discussion in the revised version.
>
> **Q1.** It is unclear why video token pruning is not also modeled with a similar learnable approach.
>
> **A6.** This asymmetry reflects the different roles of the two modalities in omni-modal compression. On the video side, redundancy mainly comes from dense spatial redundancy and local temporal redundancy, which can often be estimated from visual information itself. On the audio side, token importance more often depends on visual anchors. Therefore, the learnable module in VGAS is introduced for video-guided audio selection, rather than for query-guided semantic selection. We also note that both STVP and VGAS are query-agnostic with respect to the user query (see **A2(b)**). We will make this design motivation more explicit in the revised version.
>
> **Q2.** How does OmniSIFT account for inter-chunk temporal redundancy?
>
> **A7.** Thank you for this question. In the current version, STVP only models temporal redundancy within each chunk (see also our response in **A2 to Reviewer 4RNK**). In the revised version, we will explicitly clarify this design.
>
> **Q3.** The authors should provide robust theoretical or empirical justification for the static temporal compression strategy for video tokens.
>
> **A8.** Please see our response in **A1**.
>
> **Q4.** In more extreme scenarios, does the sequential barrier in OmniSIFT introduce a performance-latency trade-off?
>
> **A9.** Omni-video token compression is not naturally modality-parallel, as video and audio are temporally and semantically correlated, and VGAS is designed to exploit this cross-modal dependency. Therefore, the sequential design in OmniSIFT is not merely an implementation artifact, but also part of the method design for omni-video compression. We will discuss it explicitly in the revised version.
>
> Thanks again for your recognition and careful comments. Hope our detailed response above can address your concerns.

---

> > ### Author Rebuttal · Reviewer_9TNa · 2026-04-01
> >
> > Sincerely, thanks for the authors' detailed response. Most of my concerns have been solved. I am happy to maintain my positive evaluation.

---

> > > ### Author Response · Authors · 2026-04-02
> > >
> > > Thank you for your timely follow-up and for recognizing the efforts in our previous response. We sincerely apologize that the limited space prevented us from articulating our ideas clearly, and please allow us to further clarify your remaining concerns below.
> > >
> > > **W1. Uniform vs. Adaptive Allocation**
> > >
> > > We fully agree that adaptive frame allocation is naturally superior for handling non-uniform video information density. In the initial rebuttal, we provided a concrete reference case by instantiating adaptive frame allocation with the VidCom$^2$-style budgeting strategy. The purpose of this experiment was simply to offer an empirical reference, rather than to draw a broad conclusion that adaptive token compression is ineffective. We sincerely apologize if our previous wording caused any misunderstanding regarding this point.
> > >
> > > To provide a more comprehensive comparison, we have evaluated this adaptive strategy across additional omni-modal benchmarks under 35% token budget:
> > >
> > > | Method | OmniVideoBench (↑) | VideoMME (↑) | video-SALMONN-2 (↓) |
> > > | :--- | :---: | :---: | :---: |
> > > | Adaptive (VidCom$^2$-style) | 35.9 | 67.3 | 54.9 |
> > > | OmniSIFT (Uniform, Ours) | 35.6 | 68.3 | 50.5 |
> > >
> > > Beyond this, OmniSIFT's core novelty lies in **modeling redundancy across heterogeneous modalities through our modality-asymmetric framework.** While we currently use a static budget for efficiency, we fully agree that exploring adaptive allocation is a valuable future direction. Moreover, in omni-modal settings, an ideal adaptive budgeting strategy would likely need to incorporate audio cues. We will explicitly clarify this scope and discuss this promising direction in our revised manuscript.
> > >
> > >
> > > **W2. Query-guided Pruning**
> > >
> > > We highly appreciate your references to these visual pruning methods (`DivPrune`, `MoB`, `MMTok`), which provide a perfect context to discuss the boundaries of different compression paradigms. Below is a structured comparison:
> > >
> > > | Method | Target Modality | Core Pruning Guidance | Contextual Integrity |
> > > | :--- | :--- | :--- | :--- |
> > > | **DivPrune** | Vision | Intra-modality Diversity | High |
> > > | **MoB/MMTok** | Vision | Text $\rightarrow$ Vision Alignment | Challenging |
> > > | **OmniSIFT** | Vision + Audio | Query-Agnostic Vis. + Vis $\rightarrow$ Audio | High |
> > >
> > > We believe that query-guided methods and our query-agnostic STVP approach each offer unique advantages, and they are essentially **orthogonal and highly complementary**. We feel that rather than treating them as mutually exclusive, a text-query-guided selector (such as `MoB` or `MMTok`) could be seamlessly applied *on top of* OmniSIFT to further refine tokens based on user prompts.
> > >
> > > During our design process, we encountered two primary challenges with query-guided pruning in omni-modal scenarios. First, these methods face inherent challenges in **multi-turn dialogues** (e.g., SparseVILA[1]), as discarding tokens irrelevant to the current query risks losing the context required for subsequent, unpredictable questions. Furthermore, we faced the "Sound Source" dilemma in **omni-modal (Video+Audio+Text)** settings: when the semantic trigger is in the audio (e.g., "What object is making that screaming sound?"), relying solely on text-to-vision similarity often erroneously discards the visual sound source because the text lacks explicit visual entities.
> > >
> > > To navigate these, OmniSIFT preserves a holistic, query-agnostic backbone to avoid premature bias. We believe designing a query-guided mechanism robust to multi-turn context and audio-visual alignment is a highly valuable future direction. We will incorporate this discussion into our revised manuscript as a promising research direction.
> > >
> > > **Q2. Inter-chunk Temporal Redundancy**
> > >
> > > We acknowledge that from an information-theoretic view, visual data contains inter-chunk redundancies. While OmniSIFT currently focuses on intra-chunk pruning to maintain a lightweight, chunk-parallel architecture, we agree that modeling inter-chunk redundancy is a valuable research direction.
> > >
> > > Our design choice is driven by the need to **preserve prefill parallelism**: OmniSIFT processes multimodal chunks ($C_1, \dots, C_K$) in parallel to minimize TTFT latency. Introducing inter-chunk modeling would necessitate sequential dependencies, effectively breaking this parallel pipeline and significantly increasing total inference latency.
> > >
> > > Beyond these practical constraints, we believe this topic deserves further exploration. Unlike well-studied silent-video scenarios, integrating audio introduces a new dimension of complexity: as an additional temporal anchor, audio may fundamentally redefine "informative" frames across chunks. We are excited about the potential of joint audio-visual temporal modeling and will include this discussion in our revised manuscript.
> > >
> > > [1] SparseVILA: Decoupling Visual Sparsity for Efficient VLM Inference. ICCV' 25
> > >
> > > Thank you again for your insightful comments. We truly hope these detailed responses address your concerns.

---

### Official Review · Reviewer_21LA · 2026-03-08

**Soundness:** 3
**Presentation:** 4
**Significance:** 3
**Originality:** 2
**Overall Recommendation:** 3
**Confidence:** 4

**Summary:**

This paper aims to explore more efficient omni-modal large language models (Omni-LLMs) via token compression. The authors propose OmniSIFT, a modality-asymmetric token compression framework, which first conducts spatial-temporal video pruning and then performs video-guided audio compression. The idea is simple, and the authors validate the proposed method on multiple omni-modal benchmarks.

**Compliance With Llm Reviewing Policy:**

Affirmed.

**Final Justification:**

Many thanks to the authors' rebuttal. The newly provided comparison with FASTAV, a previous audio-visual compression method, and the evaluation on other LLM backbone such as Qwen3-Omni is great. But, I am afraid that the explanations on the method's technical novelty is not quite strong. It is arguable that the claimed asymmetric integration design would be a truely novel paradigm in the compression field. Moreover, the performance gap between OmniSIFT and `Direct_attention' is marginal, rasing concerns of the more complex compression design adopted in the framework. I did not see quite strong evidence in the second-round authors' reply. Although my rating may not affect the final decision, I finally keep the rating at Weak Reject. This rating also takes into account the quality of the other papers in my batch.

**Key Questions For Authors:**

see weaknesses

**Limitations:**

No. There are no discussions on the proposed method's limitations.

**Strengths And Weaknesses:**

# Strengths
- The paper is well-written and well-organized. The proposed method is clearly elaborated.
- The proposed method is validated on multiple widely used omni-modal benchmarks, and the experimental results verify the superiority of the proposed method in both general performance and efficiency.
- The discussion on efficiency-related metrics, including parameters, FLOPs, latency, and memory, is good.

# Weaknesses
- Concern about the method's novelty. 1) The authors classify the proposed method, OmniSIFT, as a modality-asymmetric compression method, different from previous methods in modality-decoupled and modality-symmetric manners. However, OmniSIFT does not utilize audio information in the first-stage video compression. The paper does not provide deeper discussion or evidence on this aspect, which may reveal the impact of audio on video compression. Moreover, the spatial and temporal video compression primarily utilize token-level cosine similarity, and the vision-guided audio compression primarily leverages audio-visual attention. These operations and ideas are widely used in prior video compression-related works and multimodal compression works, so I am afraid that they do not provide sufficiently new insights. 2) I also suggest that the authors conduct a more comprehensive literature review on omni-modal or audio-visual LLM compression, such as FASTAV: EFFICIENT TOKEN PRUNING FOR AUDIO-VISUAL LARGE LANGUAGE MODEL INFERENCE, and provide corresponding experimental comparisons if possible.
- Suggestions on the experiments. 1) All experiments use Qwen2.5-Omni backbones, limiting evidence of backbone-agnostic generalization. Experiments on different backbones would make the results more comprehensive. 2) The video-guided audio selector utilizes audio-visual attention to update the audio features and employs an MLP layer to obtain the audio saliency score. What if directly leveraging audio-visual attention as guidance for audio token pruning, or obtaining the saliency score by first concatenating audio-visual features followed by an MLP? 3) Figure 3 and Equation 5 seem to indicate that spatial-level video compression and temporal-level video compression are performed in parallel. If these two mechanisms can also be operated sequentially, what would be the effect? And how can the conflicts between these two mechanisms be resolved? For example, one token in a frame may receive a low saliency score in spatial-level compression but may be important during temporal-level frame comparison.
- It is good to see that Figure 6 provides some visualization examples of the pruned tokens. More analysis on the quality of the token compression would be appreciated.

---

> ### Author Rebuttal · Authors · 2026-03-30
>
> **Response to Reviewer 21LA**
>
> Thank you for your careful review and constructive feedback. Below, we provide point-by-point responses to your comments.
>
> **W1(a).** OmniSIFT does not use audio in the first-stage video compression, and the paper does not sufficiently discuss this design choice.
>
> **A1(a).** Our design is motivated by two key points. (1) The asymmetric treatment of video and audio comes from their different structural characteristics (see our response in **Reviewer 4RNK A1(a)**). (2) OmniSIFT is designed for token compression, so the focus here is on redundancy estimation rather than task-specific semantic relevance.
>
> In joint audio-visual reasoning, audio can certainly provide complementary semantics, especially for query-conditioned understanding. For example, in questions such as “what scene appears when a certain sound occurs?”, audio may help identify which visual content is more relevant to the task. However, this is different from deciding whether a video token is redundant. STVP is intended to remove visual redundancy, which is largely determined by visual self-similarity. By contrast, audio-token saliency benefits more directly from visual grounding. We will clarify these two points more explicitly in the revised version.
>
> **W1(b).** The local operations in OmniSIFT are related to prior video and multimodal compression works, making the novelty less clear.
>
> **A1(b).** OmniSIFT does not intend to propose new standalone visual token compression techniques. Rather, it introduces a modality-asymmetric compression method for omni-modal inputs. (see also our response in **A2 to Reviewer w67Y**)
>
> **W1(c).** More comprehensive literature review on omni-modal or LLM compression, such as FASTAV: EFFICIENT TOKEN PRUNING FOR AUDIO-VISUAL LARGE LANGUAGE MODEL INFERENCE.
>
> **A1(c).** We reproduced FASTAV and evaluated it on DailyOmni and WorldSense. Following the original paper, we use its two-stage pruning strategy: 50% global pruning at a middle decoder layer, followed by 20% fine pruning in subsequent layers. The results are as follows:
>
> | Method | DailyOmni | WorldSense |
> |:------:|:---------:|:----------:|
> | OmniSIFT(35% tokens)| 73.2 | 50.0 |
> | FASTAV | 59.4 | 43.3 |
>
> These results further support the effectiveness of OmniSIFT under a similar token budget.
>
> **W2(a).** Experiments on different backbones would make the results more comprehensive.
>
> **A2(a).** We evaluate OmniSIFT on Qwen3-Omni. Due to computational constraints, the experiment is conducted under a lightweight LoRA setting.
>
> | Backbone / Method | DailyOmni | WorldSense | DailyOmni Prefill Latency / GPU Memory | WorldSense Prefill Latency / GPU Memory |
> |:-----------------:|:---------:|:----------:|:--------------------------------------:|:---------------------------------------:|
> | Qwen3-Omni (Full Tokens) | 70.8 | 50.2 | 2.22 / 63.12 GB | 2.97 / 72.18 GB |
> | Qwen3-Omni + OmniSIFT (LoRA, 35%) | 70.5 | 48.8 | 2.16 / 60.85 GB | 2.48 / 68.56 GB |
>
> These results show that OmniSIFT generalizes to another omni-modal backbone with only minor performance drop.
>
> **W2(b).** More baseline designs for audio token compression.
>
> **A2(b).** We implement the two baselines: **Direct\_attention**, which uses audio-visual attention for audio token pruning, and **Concate\_av**, which concatenates audio-visual features. Both variants were trained under the same setting as OmniSIFT. Under the 35% token retention budget, we evaluated them on DailyOmni and WorldSense:
>
> | Method | DailyOmni | WorldSense |
> |:------:|:---------:|:----------:|
> | OmniSIFT | 73.2 | 50.0 |
> | Direct_attention | 72.4 | 49.7 |
> | Concate_av | 71.8 | 49.4 |
>
> These results show that the VGAS design achieves better overall performance under the same token budget.
>
> **W2(c)** If the two components in STVP can also be operated sequentially, what would be the effect?
>
> **A2(c).** Our current STVP is intentionally designed as a lightweight approximation. Although a joint compression scheme may be more complete in theory, it would introduce substantially higher computation for both saliency estimation and the combination of the two signals. Therefore, our STVP separately captures spatial redundancy and local temporal redundancy under a fixed budget. We will clarify this design trade-off more explicitly in the revised version and discuss joint modeling as a valuable future direction.
>
> **W3.** More analysis on the quality of the token compression would be appreciated.
>
> **A3.** Due to the limited space in the rebuttal, we are unable to include additional figures here, but we have conducted case-level analysis on the DailyOmni benchmark to better examine the behavior of OmniSIFT under compression(see our response in **A1(b) to Reviewer 4RNK**). We will add more visualization examples and related discussion in the revised version.
>
> Thank you again for your careful review and recognition of our contribution. We hope the detailed responses above help address your concerns.

---

> > ### Author Rebuttal · Reviewer_21LA · 2026-04-01
> >
> > Many thanks to the authors' rebuttal. The newly provided comparison with FASTAV, a previous audio-visual compression method, and the evaluation on other LLM backbone such as Qwen3-Omni is great. But, I am afraid that the explanations on the method's technical novelty is not quite strong. It is arguable that the claimed asymmetric integration design would be a truely novel paradigm in the compression field. Moreover, the performance gap between OmniSIFT and `Direct_attention' is marginal, rasing concerns of the more complex compression design adopted in the framework.

---

> > > ### Author Response · Authors · 2026-04-01
> > >
> > > Thank you for the follow-up and for recognizing the value of the additional experiments, including FASTAV, the extra VGAS baselines, and the Qwen3-Omni results. Regarding the comparison between VGAS and `Direct_attention`, as well as the novelty of OmniSIFT, we would like to provide the following clarification.
> > >
> > > Regarding the comparison between VGAS and `Direct_attention`, we agree that the performance gap is not large, and this is in fact expected. `Direct_attention` shares the same core intuition as OmniSIFT on the audio side, namely that audio token compression should be guided by the retained visual tokens. In this sense, **the `Direct_attention` baseline helps validate the effectiveness of this key video-guided audio compression mechanism**. The role of the MLP score head in VGAS is to provide a cleaner and more deployment-friendly scoring interface, instead of directly relying on attention weights for pruning. This design is more compatible with efficient attention implementations and adds only marginal overhead relative to the cross-attention itself.
> > >
> > > More importantly, we would like to clarify that the main novelty of OmniSIFT lies in **introducing and validating a modality-asymmetric compression paradigm for omni-modal LLMs**. Compared with prior video token compression works, omni-modal compression is not merely about handling more tokens, but about modeling two heterogeneous modalities within a unified compression framework. A straightforward extension would be modality-decoupled compression, i.e., compressing video and audio independently, while OmniZip represents a modality-symmetric alternative that treats the two modalities more equally through temporal alignment.
> > >
> > > In contrast, the key idea of OmniSIFT is that audio and video should not be compressed symmetrically in omni-modal settings. Motivated by prior literature on audio-visual asymmetry and human perception, we argue that the two modalities have inherently different redundancy structures. On the video side, redundancy is largely governed by self-redundancy within the visual stream, including spatial similarity and local temporal overlap. On the audio side, however, token saliency is more context-dependent and should be judged with visual anchors. Based on this view, OmniSIFT first performs self-redundancy modeling on the video stream, and then uses the retained key video tokens as anchors to guide audio token compression.
> > >
> > > We believe this is the main methodological contribution of our work: **proposing a new way to model redundancy across heterogeneous modalities in omni-modal compression**. More broadly, this perspective suggests that in complex omni-modal scenarios, video and audio should not always be treated as equally informative streams, and that audio may often play a complementary role conditioned on visual context. We will further clarify this point in the revised version.
> > >
> > > Thanks again for your review. We hope our response helps clarify and address your concern.

---

### Official Review · Reviewer_4RNK · 2026-03-11

**Soundness:** 3
**Presentation:** 2
**Significance:** 3
**Originality:** 3
**Overall Recommendation:** 5
**Confidence:** 3

**Summary:**

The paper proposes OmniSIFT, an efficient token compression framework for audio–video omni-modal large language models. The method is based on a modality-asymmetric design, which prioritizes visual processing and uses the resulting visual representations to guide audio token selection. Specifically, the framework first prunes spatial and temporal redundancy in video tokens to obtain a compact set of visual anchors, and then uses these anchors to retain only the audio tokens that are most relevant to the scene. Experiments on Qwen2.5-Omni-7B show that the approach significantly reduces computation, latency, and memory usage, while outperforming previous compression methods such as OmniZip and DyCoke across multiple benchmarks, and even surpassing the full-token model on several tasks.

**Compliance With Llm Reviewing Policy:**

Affirmed.

**Final Justification:**

The paper proposes a vision-dominant modality-asymmetric compression approach. Its design is conceptually clear and intuitively designed, empirically validating its key design choices and offering a lightweight yet effective strategy for multimodal representation compression. The rebuttal adequately addressed my remaining concerns, therefore I decide to raise my score to 5.

**Key Questions For Authors:**

(1). It would be helpful to clarify the scope of the statement that “visual redundancy can typically be resolved using visual cues alone,” e.g., whether it originates from purely visual settings or also applies to multimodal scenarios, and whether it still holds in speech-dominant or visually weak videos (same as Weak-1).
(2). In Figure 4 (left), there is a phenomenon that deserves further explanation: when the video compression ratio of OmniSIFT is around 0.6, the model’s accuracy instead increases. This trend appears somewhat unusual. I would appreciate it if the authors could provide further explanation or analysis of this behavior.

**Limitations:**

Yes

**Strengths And Weaknesses:**

Strength：
The paper presents a clear conceptual design in proposing a modality-asymmetric, vision-dominant compression principle. Specifically, the method first compresses visual tokens and then uses the retained visual information to guide audio token selection. Compared with existing approaches, this design introduces a relatively clear paradigm distinction at the methodological level and provides an intuitive perspective for multimodal token compression.
The paper also provides some analysis at the mechanism level. Through visualizing the sparsity of deep-layer LLM attention and conducting ablations on different selector depths, the authors show that the original multimodal representations contain substantial redundant tokens and that a relatively shallow VGAS module is sufficient to capture key cross-modal correlations. These analyses provide empirical support for several design choices in the proposed framework.
The outlier-based strategy used in STVP for pruning video frame tokens is also an interesting design choice. By measuring the deviation of patch representations relative to global frame representations or adjacent frames, the method identifies visually informative regions in a simple and intuitive way, providing a lightweight heuristic for estimating the information contribution of visual tokens.

Weakness：
(1). The applicability of the modality-asymmetric assumption is not sufficiently discussed. The paper motivates this design with the statement around line 101: “Visual redundancy can typically be resolved using visual cues alone, whereas the saliency of audio signals depends on whether the visual scene provides a semantic anchor.” While this claim suggests that visual redundancy can be addressed using visual cues alone, the paper does not clearly specify whether this notion of “visual redundancy” refers to purely visual scenarios or to joint audio–visual settings. This distinction remains somewhat unclear.
    (1.1) The overall design of the method relies on first processing visual tokens independently and then using visual signals to guide audio selection. If the cited observation mainly originates from single-modality visual settings rather than multimodal contexts, the theoretical motivation for this design may require further justification. In other words, if the assumption about “visual redundancy” does not naturally extend to multimodal scenarios, the conceptual motivation of the paper could be somewhat weakened.
    (1.2) In addition, the paper lacks systematic discussion of scenarios where audio may be more informative, such as speech-centric videos, visually noisy or low-information videos, or screen recordings. In these cases, visual signals may not serve as the most reliable modality anchor. The absence of experiments, failure case analyses, or related discussions in such settings makes it difficult to assess the generality of the modality-asymmetric paradigm and may give the impression that OmniSIFT is primarily optimized for specific types of multimodal scenarios.
(2). The paper states that the STVP module removes both intra-frame and inter-frame redundancy (Sec.3.3). However, based on the implementation details, the method mainly addresses intra-frame redundancy through spatial saliency within a frame, while the temporal saliency component simply computes cosine differences between corresponding patches of two consecutive frames and performs Top-K selection. This design appears closer to a simple frame-difference heuristic rather than an explicit modeling of inter-frame redundancy or overlap. As a result, the current description may be somewhat unclear. For example, it is not discussed whether redundancy across frames from adjacent chunks (e.g., the F2 frame of the previous chunk and the F1 frame of the next chunk) is considered. Readers might expect a clearer cross-frame redundancy modeling mechanism or a more systematic temporal compression strategy. Clarifying how STVP handles inter-frame redundancy, or more precisely describing the scope and assumptions of the module, would help reduce potential ambiguity.
(3). The order of figure references in the text appears somewhat inconsistent (e.g., Fig. 2 is mentioned before Fig. 1), which may cause minor disruption to the narrative flow. The authors may consider adjusting the citation order to maintain clearer and more consistent presentation.

---

> ### Author Rebuttal · Authors · 2026-03-29
>
> **Response to Reviewer 4RNK**
>
> Thank you for your careful review and constructive feedback. Below, we provide point-by-point responses to your comments.
>
> **W1(a).** The notion of “visual redundancy” is grounded in purely visual settings or in joint audio-visual settings?
>
> **A1(a).** Our notion of “visual redundancy” here refers to the video side within a joint audio-visual setting, rather than a purely visual scenario. Our point is that, even in joint audio-visual inputs, redundancy on the video side is still largely governed by the internal structure of the visual stream itself. This type of redundancy is largely scene-intrinsic: in both video-only and omni-modal settings, spatially homogeneous regions and repeated background patches remain forms of redundancy that can often be estimated from visual information alone. Based on this, STVP estimates redundancy from the visual side, while VGAS handles the more context-dependent audio saliency through cross-modal guidance. We will revise the paper to make this scope and motivation more explicit.
>
> **W1(b).** The absence of experiments, failure case analyses, or related discussions in such settings makes it difficult to assess the generality of the modality-asymmetric paradigm.
>
> **A1(b).** To further validate the applicability of the modality-asymmetric design in joint audio-visual settings, we performed an additional case-level comparison across Base, OmniZip, and OmniSIFT on the DailyOmni benchmark.
>
> The analysis shows that the main limitation is shared by compressed omni-modal models. In particular, the most challenging cases are those requiring fine-grained event ordering or precise audio-visual synchronization, where highly localized evidence is more vulnerable to token compression. Consistently, the dominant error categories are Event Sequence and AV Event Alignment. For example, in one AV Event Alignment case, the question asks which audio segment corresponds to a visual explanation of a technical diagram. The Base model correctly identifies the temporally aligned audio segment, while both OmniZip and OmniSIFT select incorrect but semantically similar alternatives. This suggests that distinguishing between closely related audio candidates requires precise moment-level alignment, which can be sensitive to token compression.
>
> At the same time, OmniSIFT remains consistently stronger than OmniZip across these categories, suggesting that the modality-asymmetric design helps alleviate this difficulty. We will add these failure analyses and a dedicated limitations discussion in the revised version.
>
> **W2.** Clarifying how STVP handles inter-frame redundancy, or more precisely describing the scope and assumptions of the module, would help reduce potential ambiguity.
>
> **A2.** In the current version, STVP only models local temporal redundancy within each chunk, rather than a more complete cross-frame or cross-chunk redundancy mechanism. The contribution of STVP is not a completely new visual redundancy estimator, but the integration of spatial and temporal redundancy reduction into an asymmetric omni-modal compression pipeline.(see our response in **A2 to Reviewer w67Y**) In addition, we intentionally keep STVP lightweight for efficiency: more adaptive or more global temporal redundancy modeling would require extra heuristic estimation and computation, while our current results do not show clear gains that justify this additional cost.(see our response in **A1 to Reviewer 9TNa** and **A2(c) to Reviewer 21LA**) We will state this scope and trade-off more explicitly in the revised version, and discuss richer temporal modeling as a future direction.
>
> **W3.** The order of figure references in the text appears somewhat inconsistent (e.g., Fig. 2 is mentioned before Fig. 1), which may cause minor disruption to the narrative flow.
>
> **A3.** We will carefully revise this in the revised version.
>
> **Q1.** It would be helpful to clarify the scope of the statement that “visual redundancy can typically be resolved using visual cues alone,”.
>
> **A4.** Please see our response in **A1(a)**.
>
>
> **Q2.** In Figure 4 (left), there is a phenomenon that deserves further explanation: when the video compression ratio of OmniSIFT is around 0.6, the model’s accuracy instead increases.
>
> **A5.** We believe this phenomenon is not entirely counter-intuitive: under a moderate compression ratio, removing some redundant tokens can sometimes improve cross-modal understanding, because it allows the model to focus more on truly discriminative cues. From this perspective, moderate token sparsification may play a role similar to information focusing. We will add a brief explanation of this behavior in the revised version.
>
> We sincerely thank you again for your insightful comments and your recognition of our contribution. We hope the detailed responses above help clarify your concerns.

---

> > ### Author Rebuttal · Reviewer_4RNK · 2026-04-04
> >
> > Thanks for your response. My concerns are adequately addressed,  I would like to raise my score.

---

> > > ### Author Response · Authors · 2026-04-04
> > >
> > > Thank you so much for your time in reviewing our rebuttal.  We are so glad to hear that your concerns have been addressed, and we will carefully incorporate your suggestions into the final version.

---

### Official Review · Reviewer_w67Y · 2026-03-12

**Soundness:** 3
**Presentation:** 3
**Significance:** 3
**Originality:** 3
**Overall Recommendation:** 5
**Confidence:** 4

**Summary:**

This paper first introduces asymmetric compression, where video tokens are first reduced, then the remaining tokens interact with audio embeddings through cross-attention layers. With such novel asymmetric compression, they present OmniSIFT. Experiments show that OmniSIFT effectively reduces the number of tokens while preserving the performance.

**Compliance With Llm Reviewing Policy:**

Affirmed.

**Final Justification:**

Thanks for the thoughtful rebuttal. The scope and the methods now look much more reasonable to me. I raised my score from 4 to 5. I hope these explanations will be incorporated into the revised version.

**Key Questions For Authors:**

Please refer to the weaknesses.

Here are additional questions.

1. Is STVP the same as existing works? The saliency estimation looks quite conventional.
2. Regarding the implementation of "w/o Spatial Component" in Figure 5, is it TopK(F, s2, 2 * n) or TopK(F, s2, n)?
3. How many chunks per video? Is this a very important hyperparameter? (i.e., too many or too few chunks lead to bad performance)

**Limitations:**

yes

**Strengths And Weaknesses:**

Strengths.

1. The idea of asymmetric compression is novel. And the method is intuitive and easy to understand.
2. The performance of OmniSiFT is good, approaching full tokens, and outperforms baselines at 7B-scale.
3. Ablation studies include structural ablation and token compression paradigm ablation.

Weaknesses.

1. Section 4.2
2. Regarding STVP, it is highly recommended to include the comparison with the compression method of previous works. If such saliency estimation is used in previous works, citations are encouraged in Section 3.
3. Lack of an ablation study of VGAS design, including the chunk size.
4. Though empirically effective, the design of VGAS is weird for several reasons. First, the vision embedding and audio embedding are not aligned before the dot product ($Q_aK_v^T$), which is not common. Considering when aligning vision tokens and text tokens in VLM, there is always a projector/connector. So it is uncommon to directly compute the dot product. Second, there is no positional embedding here; the sequential information is totally lost. Third, since $\mathbf H_a^{(t)}$ is always a linear combination of $\{\mathbf V_v\}$, so it would fail when all $\mathbf V_v$ are not informative. For a concrete example, if the video clip is always visually black, but the audio is content-rich.

---

> ### Author Rebuttal · Authors · 2026-03-29
>
> **Response to Reviewer w67Y**
>
> Thank you for your recognition of our work and for raising several insightful questions. Below, we provide point-by-point responses to your comments.
>
> **W1.** Section 4.2
>
> **A1.** We will fix this issue in the revised version.
>
> **W2.** The connection between STVP and prior compression methods is not sufficiently discussed.
>
> **A2.** The contribution of STVP is not a completely new visual saliency estimator, but the integration of spatial and temporal redundancy reduction into an asymmetric omni-modal compression pipeline. Related works such as VidCom$^2$[1] and TimeChat-Online[2] are already cited in the paper. In the revised manuscript, we will clarify this relation in Section 3 and better distinguish our asymmetric integration design from prior single-focus compression methods. Specifically, VidCom$^2$ performs spatial redundancy estimation on each frame, while TimeChat-Online performs temporal redundancy estimation on each frame. In contrast, our ablation results in Figure 5 show that both the spatial and temporal components in STVP are necessary, and neither can be omitted.
>
> **W3.** Lack of an ablation study of VGAS design, including the chunk size.
>
> **A3.** We have already included several ablations on VGAS in Section 4.4, including: (1) audio intra-modal self-attention, (2) random compression, (3) an SSM-based design, and (4) different numbers of Transformer layers. These results show that the current VGAS design is not arbitrary.
>
> For chunk size, we agree that it is an important factor worth further study. In the current version, we use a 2-second chunk as a practical local alignment unit for audio and video, consistent with the default granularity of Qwen2.5-Omni. Our intuition is that longer chunks would introduce larger temporal misalignment, while shorter chunks would increase the associated overhead. We will clarify this design motivation and limitation in the revised version.
>
> **W4(a).** The vision embedding and audio embedding are not aligned before the dot product $(Q_a K_v^T)$, which is not common.
>
> **A4(a).** OmniSIFT is not applied to the raw audio and visual encoder features, but to the multimodal tokens after the multimodal projector. Thus, the audio tokens and video tokens entering VGAS have already been mapped into the same representation space. We will clarify this more clearly in Section 3 to avoid ambiguity.
>
> **W4(b).** There is no positional embedding here; the sequential information is totally lost.
>
> **A4(b).** Introducing positional embedding would be more helpful for modeling temporal relations. In the current design, we mainly mitigate the loss of sequential information by following the 2-second chunk granularity of Qwen2.5-Omni (see our response in **A3**), where audio tokens and video tokens within each chunk are naturally aligned. Therefore, the role of VGAS is more focused on saliency estimation, rather than performing temporal alignment. We will clarify this design in the revised version.
>
> **W4(c).** Since $H_a^{(t)}$ is always a linear combination of $V_v$, so it would fail when all $V_v$ are not informative.
>
> **A4(c).** Thank you for pointing this out. In practice, VGAS adopts a standard residual structure, i.e.,
> $
> x = \mathrm{LayerNorm}(\mathrm{CrossAttn}(a, v, v) + a),
> $
> If all projected video tokens are nearly identical, then the cross-attention branch degenerates into a shared constant bias
> $
> x \approx \mathrm{LayerNorm}(a + c),
> $
> As a result, the selector smoothly degenerates into one that relies primarily on the audio features themselves, while still remaining capable of effective audio token compression. We will clarify this implementation detail in the revised version.
>
> **Q1.** Is STVP the same as existing works?
>
> **A5.** Please see our response in **A2**.
>
> **Q2.** Regarding the implementation of "w/o Spatial Component" in Figure 5, is it $TopK(F, s_2, 2n)$ or $TopK(F, s_2, n)$?
>
> **A6.** In the “w/o Spatial Component” setting, we select Top-$n$ from all tokens $F$ in a frame according to the temporal score $s_2$. This keeps the number of visual tokens the same as in the full STVP setting.
>
> **Q3.** How many chunks per video? Is this a very important hyperparameter?
>
> **A7.** In our implementation, we use one chunk every 2 seconds, so the number of chunks for a video is $\lceil T/2 \rceil$, where $T$ is the video duration. We consider chunk size a meaningful hyperparameter, although we have not yet conducted a systematic ablation on it (see our response in **A3**).
>
> [1] Liu X, Wang Y, Ma J, et al. Video Compression Commander: Plug-and-Play Inference Acceleration for Video Large Language Models. EMNLP'25.
>
> [2] Yao L, Li Y, Wei Y, et al. TimeChat-Online: 80% Visual Tokens Are Naturally Redundant in Streaming Videos. ACM MM'25.
>
> Thanks again for your recognition of our contribution and for your insightful comments. We hope our detailed responses above address your concerns.

---

> > ### Author Rebuttal · Reviewer_w67Y · 2026-03-31
> >
> > To A6: I think the full STVP uses 2n tokens? Since Z = [F1;F2] (Line 178), F1 and F2 both have n tokens. So this ablation may not be fair enough because full STVP has two times number of tokens as "w/o Spatial Component".

---

> > > ### Author Response · Authors · 2026-04-01
> > >
> > > Thank you for the follow-up. We apologize that our previous response did not explain this implementation detail clearly enough.
> > >
> > > More specifically, in our implementation, token compression is controlled by the compression ratio $\rho_v$ (equivalently, the retention ratio $\alpha_v = 1-\rho_v$). In the full STVP setting, we apply visual compression with ratio $\rho_v$ to both $F_1$ and $F_2$. In the w/o Spatial Component ablation, due to the nature of temporal redundancy estimation, we **apply compression with ratio $\rho_v$ only to $F_2$, while keeping $F_1$ unchanged**. In the w/o Temporal Component ablation, we apply compression with ratio $\rho_v$ to both $F_1$ and $F_2$.
> > >
> > > We also note that it is unnatural to force the w/o Spatial Component ablation to match the exact token budget of the full STVP by applying a much stronger compression ratio (e.g., $2\rho_v$) only to $F_2$, because this would excessively compress the later frame. Therefore, this ablation is intended to reflect the effect of removing the corresponding scoring component under our current implementation. We will clarify this implementation detail explicitly in the revised version, so that the ablation setting and its limitation are unambiguous.
> > >
> > > Thanks again for your careful review. We hope this response helps resolve your concern.

---

### Decision · Program_Chairs · 2026-04-30

**Decision:**

Accept (regular)

**Comment:**

Final rating: 5: Accept / 5: Accept / 4: Weak Accept / 3: Weak Reject

The paper proposes OmniSIFT, a modality-asymmetric token compression framework for Omni-modal LLMs that prioritizes visual processing to guide audio token selection. Reviewers recognized the "vision-dominant" paradigm as an insightful and fresh perspective for multimodal compression. They appreciated the method's efficiency, specifically its compatibility with FlashAttention and its ability to maintain high performance while using only 25% of the original tokens. Initial concerns focused on the technical novelty of individual components, generalizability across different backbones, and the effectiveness of fixed temporal budget allocation compared to adaptive strategies.

Following the rebuttal, three reviewers (w67Y, 4RNK, 9TNa) raised or maintained positive scores. The authors successfully addressed concerns through extensive new experiments, including evaluating OmniSIFT on a different backbone (Qwen3-Omni), providing superior comparisons against the FASTAV baseline, and implementing adaptive budget variants that validated the current efficiency-driven design. One reviewer (21LA) maintained a Weak Reject, primarily questioning the technical novelty of the asymmetric integration. However, the technical evidence provided during the discussion and the majority consensus highlight the paradigm's significance.

The AC recommends acceptance based on the technical significance of the proposed modality-asymmetric compression paradigm, while also taking the constructive author-reviewer discussion into account.